# Antibacterial Effects of Flavonoids and Their Structure-Activity Relationship Study: A Comparative Interpretation

**DOI:** 10.3390/molecules27041149

**Published:** 2022-02-09

**Authors:** Nur Farisya Shamsudin, Qamar Uddin Ahmed, Syed Mahmood, Syed Adnan Ali Shah, Alfi Khatib, Sayeed Mukhtar, Meshari A. Alsharif, Humaira Parveen, Zainul Amiruddin Zakaria

**Affiliations:** 1Drug Design and Synthesis Research Group, Department of Pharmaceutical Chemistry, Kulliyyah of Pharmacy, International Islamic University Malaysia, Kuantan 25200, Pahang D. M., Malaysia; farisya.shamsudin@live.iium.edu.my (N.F.S.); alfikhatib@iium.edu.my (A.K.); 2Department of Pharmaceutical Technology, Faculty of Pharmacy, Universiti Malaya, Kuala Lumpur 50603, Malaysia; syedmahmood@um.edu.my; 3Department of Pharmaceutical Engineering, Faculty of Engineering Technology (Chemical), Gambang Campus, Universiti Malaysia Pahang (UMP), Kuantan 26300, Pahang D. M., Malaysia; 4Faculty of Pharmacy, Universiti Teknologi MARA Cawangan Selangor Kampus Puncak Alam, Bandar Puncak Alam 42300, Selangor D. E., Malaysia; syedadnan@uitm.edu.my; 5Atta-ur-Rahman Institute for Natural Product Discovery (AuRIns), Universiti Teknologi MARA Cawangan Selangor Kampus Puncak Alam, Bandar Puncak Alam 42300, Selangor D. E., Malaysia; 6Department of Chemistry, Faculty of Science, University of Tabuk, Tabuk 71491, Saudi Arabia; sayeed_mukhtar@hotmail.com (S.M.); h.nabi@ut.edu.sa (H.P.); 7Chemistry Department, Faculty of Applied Science, Umm Al-Qura University, Makkah 24230, Saudi Arabia; Maasharif@uqu.edu.sa; 8Department of Biomedical Sciences, Faculty of Medicine and Health Sciences, Universiti Malaysia Sabah, Jalan UMS, Kota Kinabalu 88400, Sabah, Malaysia

**Keywords:** flavonoids, antibacterial effects, structure activity relationship studies, natural products, pharmacophores

## Abstract

According to the latest report released by the World Health Organization, bacterial resistance to well-known and widely available antibacterial drugs has become a significant and severe global health concern and a grim challenge to tackle in order to cure infections associated with multidrug-resistant pathogenic microorganisms efficiently. Consequently, various strategies have been orchestrated to cure the severe complications related to multidrug-resistant bacteria effectively. Some approaches involved the retardation of biofilm formation and multidrug-resistance pumps in bacteria as well as the discovery of new antimicrobial agents demonstrating different mechanisms of action. In this regard, natural products namely alkaloids, terpenoids, steroids, anthraquinone, flavonoids, saponins, tannins, etc., have been suggested to tackle the multidrug-resistant bacterial strains owing to their versatile pharmacological effects. Amongst these, flavonoids, also known as polyphenolic compounds, have been widely evaluated for their antibacterial property due to their tendency to retard the growth of a wide range of pathogenic microorganisms, including multidrug-resistant bacteria. The hydroxylation of C5, C7, C3′, and C4′; and geranylation or prenylation at C6 have been extensively studied to increase bacterial inhibition of flavonoids. On the other hand, methoxylation at C3′ and C5 has been reported to decrease flavonoids’ antibacterial action. Hence, the latest information on the antibacterial activity of flavonoids is summarized in this review, with particular attention to the structure–activity relationship of this broad class of natural compounds to discover safe and potent antibacterial agents as natural products.

## 1. Introduction

One of the immense public health challenges, at the moment, is the increasing antibiotic resistance [1,2,3,4,5,6,7]. Centers for Disease Control and Prevention (CDC) reported that more than two million cases of antibiotic-resistant infections occur each year in the United States, and at least 35,000 people perish owing to the same deleterious infections associated with the pathogenic microorganisms [8]. It was also discovered that novel resistance mechanisms are arising and spreading internationally, putting the ability to effectively combat prevalent infectious diseases at risk [9,10,11,12,13,14]. As antibiotics become less efficient, a rising number of infections, including pneumonia [15,16,17,18,19], tuberculosis [20,21,22,23], blood poisoning [24,25,26,27], gonorrhea [28,29,30,31,32,33,34], and foodborne diseases [35,36,37,38,39] have become more complex, if not, impossible to cure [7,40]. Furthermore, this problem of antibiotic resistance leads to higher medical expenditure, lengthier hospitalization, and increased death [6,7,40,41,42,43]. These problems prompted researchers to seek novel antibacterial treatments with greater potency against multi-resistant bacteria. Nowadays, there is a growing interest in generating natural products (NPs) antibacterial treatments to combat numerous complicated infections associated with multidrug-resistant pathogenic microorganisms. Flavonoids are one of a wide class of NPs that have long been recognized as critical natural compounds in the prevention and treatment of a number of diseases including infectious diseases related to a wide range of pathogenic bacteria [44,45,46,47,48,49,50].

Flavonoids, the most important class of phenolic compounds, are secondary metabolites produced by plants and are found in a non-glycosylated form (aglycone) or glycosidic form (attached to a sugar molecule, also known as glycone) [44,46,51,52,53]. All flavonoids display a standard structure of two phenyl rings (A and B) connected to a heterocyclic ring (C, the ring containing the embedded oxygen). This carbon structure can be abbreviated as C6-C3-C6. These compounds frequently display hydroxylation at positions 5 and 7 at A ring and oxidation at the 3′,4′ or 3′,4′,5′ positions at B ring due to their biosynthesis routes [46,54,55]. Flavonoids are frequently found in nature and have a significant chemical diversity mainly because of the changes in B- and C-ring locations, degree of hydroxylation, oxidation, and saturation of ring C [51,56,57]. Flavonoids are classified according to their chemical structures into chalcones, flavanones, flavanonols, flavones, flavonols, isoflavones, flavan-3-ols (catechins), and anthocyanidins, as shown in Figure 1. The numbering system of a basic flavonoid structure is also illustrated in Figure 2. The presence of a ketonic functional group at position 4, a double bond between carbon atoms 2 and 3 (C2=C3 double bond), and the presence of a hydroxyl group at position 3 (3-OH) in the ring C are the primary factors that precisely determine flavonoids subdivision.

Flavonoids have been identified as polyphenolic compounds capable of exerting antibacterial activities via various mechanisms of action. According to multiple research studies, flavonoids can suppress nucleic acid synthesis, cytoplasmic membrane function, and energy metabolism [58,59,60,61,62,63]. Flavonoids have also been found to reduce adhesion and biofilm formation, porin on the cell membrane, membrane permeability, and pathogenicity, all of which are crucial for bacterial growth [58,60,61,63]. Furthermore, certain flavonoids have been reported to reverse antibiotic resistance and improve the efficacy of the present antibiotics [60,61,64,65,66,67]. As a result, the development and deployment of flavonoid-based medications could be a potential method for antibiotic-resistant infections. This review is an attempt to increase our knowledge and understanding of flavonoids’ antibacterial effects, emphasizing on their structure-activity relationship (SAR). 

## 2. Literature Sources and Search Strategy

An online literature search utilizing phrases such as ‘structure–activity relationship study of flavonoids as antibacterial agent’; ‘flavonoids as antibacterial agent’; ‘antibacterial effect of flavonoids’; ‘in vitro antibacterial activity or property of flavonoids’; ‘SAR of flavonoids as antibacterial agent’ yielded information about antibacterial properties of flavonoids and their SAR analysis. The online databases used were Google Scholar, Wiley Online, Springer Link, Science Direct, PubMed, and the International Islamic University Malaysia (IIUM) Discovery Service. The online search was tailored between the years 2000 to 2021, giving almost 400 easily accessible studies on the SAR of flavonoids as antibacterials that were predominantly published in English. The retrieved data was critically examined by looking for descriptions of flavonoids’ antibacterial activities and their SAR. Additional data was gathered by reviewing and analyzing the references cited in the selected articles. As a result, the current review accounts for previous and most recent information on flavonoids’ antibacterial effects, focusing on their SAR.

## 3. Discussion

### Anti-Bacterial Activity of Flavonoids

Flavonoids have been reported to exert varying degrees of antibacterial activity. Otsuka et al. (2008) evaluated the antibacterial potential of laurel leaves (*Laurus nobilis*) and consequently isolated two anti-methicillin resistant *Staphylococcus aureus* (MRSA) agents, namely kaempferol 3-*O*-α-l-(2‴,4‴-di-*E*-*p*-coumaroyl)-rhamnoside (**1a**) and kaempferol 3-*O*-α-l-(2″-*Z*-*p*-coumaroyl-4‴-*E*-*p*-coumaroyl)-rhamnoside (**1b**). Based on Table 1, both compounds outperformed antibacterial agents such as tetracycline, erythromycin, norfloxacin, ciprofloxacin, and oxacillin in terms of anti-MRSA activity with minimum inhibitory concentration (MIC) of 0.5–2.0 μg/mL. SAR studies for both compounds revealed that 5,7-dihydroxylation of the ring A and 4′-hydroxylation of the ring B are important structural features (pharmacophores) for eliciting an in vitro anti-MRSA activity [68]. Furthermore, substituting 3-O-acyl chains with C8 and C10 in the ring C increased the anti-staphylococcal action of flavonoids belonging to the flavan-3-ol class [69]. Both compounds possessed 3-O-acylated rhamnose. Both compounds include a ten-carbons acyl (coumaroyl) group and showed that this portion would likewise contribute to the strong anti-MRSA activity of **1a** and **1b**. Figure 3 shows the summary of SAR analysis of tested flavonoids [70].

Moreover, Šmejkal et al. (2008) successfully isolated six C6-geranylflavonoids (as shown in Table 2 and Figure 4) from *Paulownia tomentosa* fruits ethanol extract and their antibacterial properties were further explored against six different bacterial strains. It was reported that all the isolated compounds showed comparable antibacterial activity when compared to commercialized drugs, viz., ciprofloxacin and nystatin. Šmejkal et al. postulated that the presence of OH at C5 further increased the action of the flavanones tested. The hydroxyl group at C5 can establish an intramolecular hydrogen bond with the carbonyl group at position C4, resulting in more electron delocalization inside the molecule. However, the introduction of methoxy groups at any R position was found to diminish the antibacterial potential of the aforementioned isolated compounds by retarding the inhibition of bacterial growth. In addition, the presence of the geranyl group at C6 increased the antibacterial activity by increasing compounds’ lipophilicity characteristics. 3′-methoxy-4′,5′-dihydroxyphenyl groups of ring B of compound 3′-*O*-methyl-5′-hydroxydiplacone (**2a**) and 3′,5′-dimethoxy-4-hydroxy substitution of compound 3′-*O*-methyl-5′-*O*-methyldiplacone (**2b**) appeared to be crucial for activity due to an increase in the molecule’s planar character [71].

Similarly, Wu et al. (2013) revealed the important structural aspect in *Escherichia coli* DNA gyrase inhibition exerted by the flavonoids, as shown in Figure 5. DNA gyrase is an essential enzyme for bacterial replication as it can relieve the strain within double-stranded DNA during DNA replication. By looking at the bioactivity results in Table 3, kaempferol with a hydroxyl group at C3′, quercetin with a hydroxyl group at C3′, and galangin without 4′-OH all demonstrated that 3′-OH decreased activity and emphasizing the importance of 4′-OH in the ring B. In addition, myricetin and quercetin, which possess hydroxyl group at C5′, suggested that 5′-OH also decreased the flavonoids’ antibacterial activity. Next, baicalein with 6-OH was less efficient as an antibacterial agent, showing that the hydroxyl group at C6 in ring A is not essential and may hinder the compound’s biological action. It was also reported that 3-OH in ring C could decrease the DNA gyrase inhibition, as shown in galangin when compared to chrysin. Apart from that, it has been observed that substituting a lipophilic functional group, such as OCH_3_, at C6 or C8 increases antibacterial activity of the resultant flavonoids [72]. Tangeritin, for example, displayed more excellent activity than 5,6,7,4′-tetramethoxyflavone, with 8-OCH_3_ in the ring A being the only structural difference. However, the substitution of 3′-OCH_3_ in ring B reduces the inhibitory effect on bacteria, as can be seen in nobiletin. Next, the authors reported that hydroxylation at C5 boosted the activity as seen in baicalein, while the methoxy group at the same position decreased activity as shown by 5,6,7,4-tetramethoxyflavone. Summary of the structural requirement as shown in Figure 5 [62].

In another similar research work performed by Wu et al. (2013), the authors evaluated the antibacterial activity of several flavonoids (viz. chrysin, kaempferol, 5,6,7,4′-tetramethoxyflavone, luteolin, baicalein, quercetin, tangeritin, daidzein, puerarin, genistin, and ononin) on *E. coli* via membrane interaction effect. The bioactivity results are displayed in Table 4. Results indicated that the hydroxyl group at position 3 in ring C was found important to manifest the antibacterial properties of flavonoids. For example, quercetin (a flavonol) showed a higher antibacterial ability than luteolin (a flavone) which lacked the OH group at position 3 compared to quercetin. It was also revealed that the presence of methoxy groups reduced the antibacterial properties of flavonoids, i.e., tangeritin (4′,5,6,7,8-pentamethoxyflavone) and 5,6,7,4′-tetramethoxyflavone showed lower antibacterial activity than the flavonoids bearing hydroxyl groups. However, the methoxy group at C8 in ring A appeared to increase the antibacterial activity. Tangeritin showed higher activity than 5,6,7,4′-tetramethoxyflavone despite the similarity in their structures except for a methoxy group at C8 position in ring A. It was also suggested that the antibacterial activity of isoflavonoids decreased with the glycosylation at C7 and C8 in the ring A, such as daidzein which exhibited the highest inhibitory activity compared to puerarin, genistin, and ononin (Figure 6) [73].

Next, Yadav et al. (2013) studied the SAR of 15 selected flavonoids and their antitubercular effect against *Mycobacterium tuberculosis*. It was reported that luteolin (MIC = 25 μg/mL) had the most strong antibacterial activity among the all flavonoids evaluated. However, quercetin, baicalein, and myricetin exhibited similar activity (MIC = 50 μg/mL) and hispidulin showed the least antibacterial effect (MIC = 100 μg/mL), while the others were found inactive. They discovered that the presence of hydroxyl groups at the positions 5, 6, and 7 in ring A such as baicalein, or the 3′,4′ positions in ring B such as luteolin, quercetin, and myricetin, or the 4′ position in 5,7-dihydroxy-6-methoxy arrangement as in hispidulin were indispensable for inhibiting the growth of *M. tuberculosis*. However, the addition of random hydroxyl substitutions did not influence the antitubercular action of the flavonoids studied in the same research work. Additionally, this SAR study also demonstrated that antitubercular activity requires the presence of two hydroxyl groups in neighboring locations. However, compounds become inactive when the hydroxyl group is methylated or glycosylated, as seen with casticin and hesperidin [74]. The overall requirements for the antitubercular activity of flavonoids are shown in Figure 7.

Ullah et al. (2006) synthesized several 4-thioflavones and 4-iminoflavones derivatives, and revealed their antibacterial properties on linear growth inhibition of *E. coli*, *B. subtilis*, *S. aureus*, *Shigella flexneri*, *Pseudomonas aeruginosa*, and *Salmonella typhi*. The structural-activity analysis revealed that 4-iminoflavones exhibited better activities compared to 4-thioflavones, due to the presence of 2,4-dinitrophenyl hydrazine moiety. Next, it was also found out that the antibacterial activity of all the derivatives of 4-thioflavone and 4-iminoflavone was increased in the presence of methoxy group at 4′-position in ring B. The same pattern of activity was also found when OCH_3_ group in ring B was replaced by NO_2_ group; however, the bacterial percentage inhibition was decreased slightly (Figure 8). Fluorine at C4′ showed remarkable inhibitory properties against *E. coli*, *B. subtilis*, *S. flexneri*, *S. typhi*, and *P. aeruginosa* but were inactive against *S. aureus*. It was discovered that methoxy groups at C3′ and C4′ of ring B can enhance the inhibition of *E. coli*, *P. aeruginosa* and *S. typhi* growth. On the other hand, NO_2_ group at C5′ lowered the inhibition properties [75].

Echeverría et al. (2017) tested eight flavonoids (three flavanones and five flavones) for their antibacterial activity against four Gram-positive (*B. cereus*, *B. subtilis*, *Bacillus coagulans*, and *S. aureus*) and four Gram-negative (*Enterobacter cloacae*, *E. coli*, *Klebsiella pneumoniae*, and *Proteus mirabilis*) bacteria. The antibacterial activities are displayed in Table 5. It can be seen that flavones were more active against Gram-positive bacterial inhibition compared to flavanones. Therefore, it can be concluded that C2=C3 double bond may be essential for inhibiting bacterial growth. In flavanones, the Gram-positive bacterial inhibition became weaker when the number of OH groups was greater as seen in naringenin and 7-*O*-methyl eriodictyol with three OH groups. In comparison, pinocembrin with two hydroxyl groups at ring A showed the lowest MIC. In addition, the authors also pointed out that the number of OH groups was not important but the position of two hydroxyl groups in the flavanones was found to be essential and important in exerting antibacterial effects. Therefore, it was stressed that the presence of two OH groups in ring A and the absence of any hydroxylation in ring B influenced good antibacterial activity, as seen in the most active flavanone, pinocembrin. In flavones, with galangin and 3-*O*-methylgalangin as the most active compound, it can be deduced that the OH groups are important for antibacterial activity. It is because galangin has three OH groups (two in ring A), followed by 3-*O*-methylgalangin with two OH in ring A in their structures. Similar to flavanones, the lack of substitutions in ring B seemed to be essential for the activity. Then, the authors also mentioned that the methylation of OH groups at ring A would decrease the antibacterial action. It was proved as 3-*O*-methylgalangin with the methylation of one OH group in ring A had weaker antibacterial activity than the galangin. Furthermore, the methylation of both OH groups at ring A (as seen in 3,7-*O*-dimethylgalangin) led to inactivity in antibacterial inhibition. Figure 9 summarizes all the above-mentioned critical structural features of the flavonoids for antibacterial activity [76].

Ávila et al. (2008) studied the SAR of chalcones (described as minor flavonoids), as aromatic ketones with two phenyl rings that are also intermediates in the synthesis of many biological compounds, including flavonoids and isoflovonoids. The closure of hydroxychalcones causes the formation of the flavonoid structure in antibacterial activity by testing 31 chalcones compounds. The bacteria used in this study were *S. aureus* ATCC 25923, *P. aeruginosa* ATCC 27853, *E. coli* ATCC 25922, and *B. cereus* ATCC 11778. The results showed that 4-hydroxyderricin (MIC ≥ 3.9 μg/mL) had the highest antibacterial activity, followed by 2,4,4′-trihydroxy-3-prenyl-3′-geranylchalcone (MIC ≥ 15.6 μg/mL) and 2′,4,4′-trihydroxy-3′-geranylchalcone (MIC ≥ 15.6 μg/mL). After comparing all 31 chalcones, several necessary structural requirements for antibacterial action were confirmed, as shown in Figure 10. Firstly, hydroxylation at C2′ seemed to be an important feature as acetylation or methylation at those positions weakened the bioactivity. Secondly, hydroxyl group at C4′ position of ring B was also one of the important characteristics as the chalcone containing the hydroxyl group at the same position became inactive once it underwent methylation. Moreover, the absence of double bonds was also found to decrease the chalcones’ activity. Additionally, the hydrophobic group such as prenyl and geranyl groups at C3 position of ring A was also found important to improve the chalconesʹ lipophilicity and facilitate its entry into the bacteria cell membrane [77].

Xu and Lee (2001) tested the antibacterial activity of 38 plant-derived flavonoids on several antibiotic-resistant bacteria such as MRSA, multidrug-resistant *Burkholderia cepacian* and vancomycin-resistant enterococci (VRE). As is depicted in Table 6, it was reported that myricetin, datiscetin, kaempferol, quercetin, and luteolin were active against MRSA. Meanwhile, only myricetin was found active on multidrug-resistant *B. cepacia* and VRE. Several structural features requirements have been discussed, as illustrated in Figure 11. Firstly, presence of polyhydroxylation may lead to higher bioactivity as seen as myricetin, datiscetin, quercetin, luteolin, and kaempferol. The authors stressed that trihydroxylation at C3′, C4′, and C5′ of ring B was important, especially against multidrug-resistant *B. cepacian* and VRE. It was also confirmed that hydroxylation at ring B was important as the absence of the hydroxyl groups led to inactivities, which was seen in pinocembrin, chrysin, galangin, kaempferide, and tamarixetin. Hydroxylation at the C3 position of ring C was also found essential for a good activity. It was also observed that the glycosylation at C3 position instead of hydroxylation in myricitrin and rutin caused inactivity. The methoxylation at C3 position also led to inactivities such as in ombuin and rhamnetin. Moreover, when quercetin and inactive catechin were compared, it was suggested that C2=C3 double bond might also be important for the manifestation of antibacterial activity [64].

Yin et al. (2004) discussed the SAR of antibacterial activity of several flavonoids isolated from *Psoralea corylifolia* seeds. The isolated flavonoids were initially tested for their antibacterial potential against two Gram-positive bacteria, namely *S. aureus* and *S. epidermidis*. Nine flavonoids, namely corylifol B, neobavaisoflavone, isobavachalcone, 7,8-dihydro-8-(4-hydroxyphenyl)-2,2-dimethyl-2H,6H-benzo[1,2-*b*:5,4-*b*′]dipyran-6-one, isoneobavaiso-flavone, bavachalcone, bavachin, bavachinin, and one novel compound (erythrinin A) exhibited significant antibacterial activity against both bacteria with an average MIC of 0.009–0.073 mM (Table 7). Isobavachalcone, bavachinin, and erythrinin A showed more potent activity with MIC values of two times lower than two well-known antibacterial agents, namely bakuchiol and magnolol. From the bioactivity, the SAR study of these flavonoids as antibacterial agents revealed that for the chalcone derivatives, prenyl group at ring A led to more active antibacterial activity as exhibited by corylifol B, isobavachalcone, and bavachalcone. It was further proved by observing 1-[2,4-dihydroxy-3-(2-hydroxy-3-methyl-3-butenyl)phenyl]-3-(4-hydroxyphenyl)-2-propen-1-one, bakuchalcone, and brosimacutin G, in which the prenyl group of ring A was either oxygenated or cyclized, the compounds became inactive. As corylifols B with an extra hydroxyl group at C3′ and C4′ had higher activity compared to bavachalcone with single C4′-OH group, it was seen that the additional hydroxylation led to the stronger antibacterial action. Next, for prenylflavone derivatives, presence of three hydroxyl groups produced more antibacterial active compounds, as shown in 7,8-dihydro-8-(4-hydroxyphenyl)-2,2-dimethyl-2H,6H-benzo[1,2-b:5,4-b′]dipyran-6-one, bavachinin, and bavachin. Furthermore, the methoxylation at C7, as shown in bavachinin, had stronger activity compared to hydroxylation at C7 which can be seen in bavachin (Figure 12) [78].

Alcaraz et al. (2000) examined the 18 flavonoids comprising chalcones, flavanones, and flavones, and studied the SAR concerning antibacterial activity against MRSA strain using quantitative analysis. Based on the quantitative analysis, several characteristics were pointed out. Firstly, the authors mentioned that chalcones portrayed higher antibacterial activity compared to flavanones and flavones. It can be said that chalcone with the presence of broken ring C may enhance the anti-MRSA action. It was reported that carbonylic group is a crucial structural feature for anti-MRSA. It was also observed that the hydroxylation in unsubstituted flavonoids increased the activity. Hence, the hydroxylation at C2′ of chalcones and C5 of flavanones and flavones led to an increase of antibacterial activity. Meanwhile, the -OCH_3_ group at the above-mentioned positions reduced the antibacterial action. Lastly, halogen introduction in unsubstituted flavonoids failed to modify the antibacterial activity (Figure 13) [79].

Xie et al. (2017) evaluated antibacterial effects of some selected and structurally alike flavonoids, particularly possessing pyrogallol functionality in different rings and performed a SAR study to discover a potent antibacterial agent. They reported that balcalein and myricetin exhibited the most substantial antibacterial effect among all the flavonoids, with MIC values ranging between 0.5–2.5 mmol/L. The structure of baicalein and myricetin were taken into account to understand their antibacterial potential from SAR perspective. It was observed that both the compounds possessed pyrogallol structure in which three hydroxyl groups were present at the adjacent carbons of the benzene ring namely, balcalein in which trihydroxylation (benzenetriols) existed in ring A at C5, C6, and C7 positions, respectively, however in the case of myricetin, positions C3′, C4′, and C5′ had hydroxyl groups in ring B. Therefore, it was deduced that pyrogallol structure is an important structural feature for the manifestation of good antibacterial activity in the case of flavonoids. Meanwhile, for flavonols and flavanones, hydroxylation at C5 and C7 of ring A also seemed important as shown in quercetin, rutin, naringenin, and hesperitin. It was also observed that flavanones (naringenin) had stronger antibacterial activity than corresponding flavones (apigenin). Hence, the saturation of C2=C3 double bond, which can be seen in flavanones, was also considered an important structural feature for antibacterial activity. The structures of all mentioned flavonoids and their related SAR for antibacterial activity can be viewed in Figure 14 [80].

Hummelova et al. (2014) studied the antibacterial activity of 15 naturally occurring isoflavones against nine Gram-positive and Gram-negative bacteria using the broth microdilution method and the antibacterial effect was expressed as MIC. It was reported that only five isoflavones showed significant antibacterial activity as depicted in Table 8. 6,7,4′-trihydroxyisoflavone (**3f**), 5,7-dihydroxy-4′-methoxyisoflavone (**3i**), 5,7,4′-trihydroxyisoflavone (**3n**), 7,3′,4′-trihydroxyisoflavone (**3g**), and 7,8,4′-trihydroxyisoflavone (**3j**) showed remarkable antibacterial activity with MIC of ≥16, ≥32, ≥64, ≥128, and ≥128 µg/mL, respectively. Results indicated that **3f** was the most active isoflavone among the tested isoflavones. The SAR study revealed that the hydroxylation at C7 and C5 seemed to be important for these isoflavones. It was also mentioned that C4′-OH group could improve the antibacterial activity of isoflavones. In addition, the most active compound, namely **3f** had hydroxylation at C7, C4′, and C6 positions. Therefore, hydroxylation at the C6 position was also found to influence the antibacterial activity. Moreover, the presence of methoxy groups in ring A or at C4ʹ led to inactivity, which was seen in several methylated isoflavones, namely 6,4′-dimethoxy-7-hydroxyisoflavone (**3a**), 6,7,4′-trimethoxyisoflavone (**3b**), 7,4′-dimethoxyisoflavone (**3c**), 5,7,4′-trimethoxyisoflavone (**3d**), 7-hydroxy-6methoxyisoflavone (**3e**), 7,4′-dimethoxy-5-hydroxyisoflavone (**3h**), 5,4′-dihydroxy-7-methoxyisoflavone (**3k**), 7,4′-dihydroxy-6-methoxyisoflavone (**3l**), and 7-hydroxy-4′-methoxyisoflavone (**3o**) (Figure 15) [81].

Feng et al. (2014) conducted the SAR studies while evaluating the antibacterial potential of chalcone (precursor of flavonoids) derivatives. It was reported that the chalcone derivatives exhibited potent antibacterial action against Gram-positive bacteria compared to other cyclized derivatives. It can be observed from Table 9 that (*E*)-3-(2-(Allyloxy)phenyl)-1(2,4-dihydroxyphenyl)prop-2-en-1-one (**4d**) with C7, C9-dihydroxyl groups and alloxy group at C2 showed the best activity with MIC of 0.39−6.25 μg/mL. **4d** displayed greater activity than (*E*)-3-(2-(Allyloxy)phenyl)-1-(4-hydroxyphenyl)prop-2-en-1-one (**4i**), with hydroxy group at C9 of ring A being the only structural difference, indicating that C9 hydroxylation is important for Gram-positive bacterial inhibition. Next, hydroxylation at C5 of ring A had been reported to reduce Gram-positive bacterial inhibition, as can be seen in (*E*)-3-(6,6-Dimethylbicyclo[3.1.1]hept-2-en-2-yl)-1-(2,4,6-trihydroxyphenyl)prop-2-en-1-one (**4c**) and (*E*)-3-(2-(Allyloxy)phenyl)-1-(2,4,6-trihydroxyphenyl)prop-2-en-1-one (**4h**) when compared to (*E*)-1-(2,4-Dihydroxyphenyl)-3-(6,6-dimethylbicyclo[3.1.1]-hept-2-en-2-yl)prop-2-en-1-one (**4a**) and **4d**. Then, lipophylic *O*-alkyl substituents, such as allyloxy group (**4d**), n-hexyloxy [(*E*)-1-(2,4-Dihydroxyphenyl)-3-(2-(hexyloxy)phenyl)prop-2-en-1-one (**4e**)] and n-octyloxy [(*E*)-1-(2,4-Dihydroxyphenyl)-3-(2-(octyloxy)phenyl)prop-2-en-1-one (**4f**)] at the C2 of ring A position were also found to improve the antibacterial activity. The authors also proposed that the free phenol hydroxyl group at C7 was important for antibacterial activity against Gram-positive bacteria because chalcone protected with *O*-methoxymethyl (MOMO) at C7 was inactive, as shown in (*E*)-3-(6,6-Dimethylbicyclo[3.1.1]hept-2-en-2-yl)-1-(2-hydroxy-4-(methoxymethoxy)phenyl)prop-2-en-1-one (**4b**). Lastly, the presence of one hydroxyl group at C5 or C9 was significant. It is because of the presence or absence of hydroxyl group at both these positions led to a weaker antibacterial activity. The summarization of the important structural characteristics for antibacterial chalcone derivatives is shown in Figure 16 [82].

Simard et al. (2016) constructed a summary of critical structural features for anti-MRSA flavonoids isolated from balsam poplar buds. The isolated flavonoids known as balsacones (also known as derivatives of dihydrocinnamoyl flavans) were tested against ten MRSA strains obtained from Chicoutimi hospital in Saguenay, Province Québec, Canada by using liquid microdilution method. The results showed that balsacone R was the most active compound with a MIC value < 10 µM against all MRSA strains. On the other hand, balsacone Q was the second most active compound as it exhibited MIC value < 10 µM on half of the MRSA strains. Figure 17 shows the crucial structural characteristics of balsacones on their anti-MRSA activity. Firstly, the presence of cinnamoyl or dihydrocinnamoyl chain at C8 (R_1_) led to stronger anti-MRSA action. Next, *p*-hydroxycinnamyl group at the C3 position (R_2_) was said to be supplemental but not essential for anti-MRSA activity. Next, the hydroxylation at C3 (R_2_) weakened the antibacterial activity compared to the unsubstitution or presence of cinnamyl group at C3. Then, it was found that the 3,4-dihydropyran ring fused with ring A at C5 and C6 (R_3_ and R_4_) without *p*-hydroxycinnamyl group at C8′ did not cause any reduction in anti-MRSA activity, but the presence of *p*-hydroxycinnamyl group at C8′ completely inactivated the compound. Next, the unsubstitution at C6 (R_4_) with the presence of *p*-hydroxycinnamyl was found to be tolerated [83].

Omosa et al. (2016) also evaluated the anti-MRSA activity of various flavonoid derivatives isolated from Kenyan plants. Three chalcone derivatives, namely 32,032,5′-dihydroxy-1′-methoxychalcone (**5a**), 1′,3′-dihydroxy-2′,5′-dimethoxychalcone (**5b**), and 1,5-diacetate-3′-methoxychalcone (**5c**), were found (as shown in Table 10) to be active against four MRSA strains tested, with MIC values ranging from 16 to 128 µg/mL. It can be seen that **5a** and **5b** were the most active compounds. However, by observing both compounds, it was revealed that hydroxylation at C6 (R_2_) was responsible for chalcone’s anti-MRSA activity. The meta orientation of the methoxy and hydroxyl groups in ring A was also accountable for chalcone’s MRSA inhibition. It can be seen that **5b** had better inhibitory action than **5a**, implying that additional oxygenation at the C1 position can enhance anti-MRSA action. Furthermore, the authors investigated the antibacterial activity of different flavanones and polymethoxylated flavones against MRSA strains. However, due to their high lipophilicity character, all aforementioned compounds demonstrated no activity against tested bacteria. Hence, hydrophilicity was proved to be a critical factor in anti-MRSA action. Then, it was substantiated by the fact that the most lipophilic flavone, namely 5,7-diacetate-3,4′-trimethoxyflavone, displayed minimal activity against certain MRSA bacteria with MIC values ranging from 128 to 256 g/mL. Figure 18 summarizes the structural requirements for the tested chalcones against MRSA strains [84].

Linden et al. (2020) performed a structural investigation on the antibacterial activity of biflavonoids derived from the fruits of the Brazilian Peppertree (*Schinus terebinthifolius* Raddi). Agathisflavone, amentoflavone (AMF), and tetrahydroamentoflavone were among the biflavonoids examined. The structure of biflavonoids studied can be seen in Figure 19. Tetrahydroamentoflavone was shown to have the best antibacterial activity against *B. subtilis* and *Staphylococcus carnosus*, with MIC ≥ 0.063 mg/mL. It implied that reduced C-ring influenced tetrahydroamentoflavone’s strong antibacterial activity. On the other hand, amentoflavone with an oxidized ring C did not impede the growth of the tested bacteria at concentrations ranging from 0.016 mg/mL to 1 mg/mL. However, at a concentration of 1 mg/mL, agathisflavone, the constitutional isomer of AMF, inhibited bacterial growth of *S. carnosus* by one log unit. As a result, dimerization and the saturated state of the ring C were shown to affect the antibacterial activity of the biflavonoids studied [85].

Kong et al. (2015) evaluated antibacterial effect of two flavonoid derivatives (flavan) against *E. coli* by microcalorimetry and chemometrics methods. The compounds were liquiritigenin (aglycone) and liquiritin (glycone), which are illustrated in Figure 20. According to the authors, the antibacterial impact of liquiritin was roughly two times stronger than the antibacterial effect of liquiritigenin. The only structural variation between these two compounds was at position C4′ of ring B, where liquiritigenin included a hydroxyl group; however, liquiritin contained a sugar moiety. It was hypothesized that substitution at C4′ of ring B is vital for imparting antibacterial action of these tested flavonoids. Furthermore, the authors speculated that the presence of four free hydroxyl groups in the sugar moiety of liquiritin might be one of the most important reasons contributing to the compound’s enhanced antibacterial action [86].

A recent study conducted by Song et al. (2021) examined the antibacterial activity of isobavachalcone against multidrug-resistant strains. According to Table 11, isobavachalcone exhibited superior bacterial inhibition compared to the commercially available drug, vancomycin. The authors then demonstrated that prenylation might be responsible for the activity because another compound from the xanthone class, known as α-mangostin, which contains a prenyl group, also exhibited remarkable inhibitory activity against bacteria. The structure of isobavachalcone is depicted in Figure 21 [65].

Bahrin et al. (2014) synthesized and tested antibacterial properties of novel sulfur-containing tricyclic flavonoid derivatives (**6a**–**h**) against *S. aureus* and *E. coli*. Table 12 demonstrates that 8-bromo-4-(4-chlorophenyl)-*N*,*N*-diethyl-4*H*-[1,3]dithiolo[4,5-*c*]chromen-2-amine (**6a**), 8-bromo-*N*,*N*-diethyl-4-(4-methoxyphenyl)-4*H*-[1,3]dithiolo[4,5-*c*]chromen-2-amine (**6b**), 4-(4-chlorophenyl)-*N*,*N*-diethyl-6,8-diiodo-4*H*-[1,3]dithiolo[4,5-*c*]chromen-2-amine (**6c**), 6,8-dibromo-4-(4-chlorophenyl)-*N*,*N*-diiethyl-4*H*-[1,3]dithiolo[4,5-*c*]chromen-2-amine (**6d**), and 4-(4-chlorophenyl)-*N*,*N*-diethyl-4H-[1,3]dithiolo[4,5-*c*]chromen-2-amine (**6e**) exhibited good antibacterial activity. The authors reported that **6a**–**e** exerted better inhibition compared to their equivalent flavanones. Therefore, it can be concluded that the third fused cycle as highlighted in Figure 22 was responsible for the antibacterial action. The authors mentioned that the fused cycle’s electrophilic atom could interact with nucleophilic moieties of bacterial wall or membrane constituents, particularly in Gram-positive bacteria, which have more nucleophilic sites than Gram-negative bacteria [87]. The statement was backed up by the fact that **6a**–**e** were more sensitive to *S. aureus* than *E. coli*. Next, it can be seen that **6b** had lower *E. coli* inhibition compared to **6a**. It could be ascribed to the substitution of a chlorine atom to methoxy group at C4′ of ring B. Besides, changing the hydrogen atom (**6a**) at C8 of ring A to iodine or bromine (**6c**,**d**) substituent found to decrease the antibacterial activity in both tested strains [88].

In the following year, Bahrin et al. (2016) synthesized another set of sulfur-containing tricyclic flavonoids. Figure 23 depicts the chemical structures of the derivatives, and Table 13 depicts their antibacterial properties. It can be observed that the larger *N*,*N*-dialkylamino group as shown in 2-(Piperidin-1-yl)-8-bromo-4-(4-chlorophenyl)-4*H*-1,3-dithiol[4,5-*c*]chromen-2-ylium tetrafluoroborate (**7c**) caused a massive reduction in antibacterial activity compared to other compounds. The authors summarized that the better substitution for antibacterial activity at R_2_ started with NEt_2_ [2-*N*,*N*-Diethylamino-8-bromo-4-(4-fluorophenyl)-4*H*-1,3-dithiol[4,5-*c*]chromen-2-ylium tetrafluoroborate (**7d**), 2-*N*,*N*-Diethylamino-8-bromo-4-(4-bromophenyl)-4*H*-1,3-dithiol[4,5-*c*]chromen-2-ylium tetrafluoroborate (**7e**), 2-*N*,*N*-Diethylamino-8-bromo-4-(4-iodophenyl)-4*H*-1,3-dithiol[4,5-*c*]chromen-2-ylium tetrafluoroborate (**7f**), 2-*N*,*N*-Diethylamino-8-bromo-4-phenyl-4*H*-1,3-dithiol[4,5-*c*]chromen-2-ylium tetrafluoroborate (**7g**), 2-*N*,*N*-Diethylamino-8-iodo-4-(4-fluorophenyl)-4*H*-1,3-dithiol[4,5-*c*]chromen-2-ylium tetrafluoroborate (**7h**), 2-*N*,*N*-Diethylamino-8-iodo-4-(4-chlorophenyl)-4*H*-1,3-dithiol[4,5-*c*]chromen-2-ylium tetrafluoroborate (**7i**), 2-*N*,*N*-Diethylamino-8-iodo-4-(4-bromophenyl)-4*H*-1,3-dithiol[4,5-*c*]chromen-2-ylium tetrafluoroborate (**7j**), 2-*N*,*N*-Diethylamino-8-iodo-4-(4-iodophenyl)-4*H*-1,3-dithiol[4,5-*c*]chromen-2-ylium tetrafluoroborate (**7k**), 2-*N*,*N*-Diethylamino-8-iodo-4-phenyl-4*H*-1,3-dithiol[4,5-*c*]chromen-2-ylium tetrafluoroborate (**7l**), 2-*N*,*N*-Diethylamino-4-phenyl-4*H*-1,3-dithiol[4,5-*c*]chromen-2-ylium tetrafluoroborate (**7m**)]>pyrrolidine [2-(Pyrrolidin-1-yl)-8-bromo-4-(4-chlorophenyl)-4*H*-1,3-dithiol[4,5-*c*]chromen-2-ylium tetrafluoroborate (**7b**)]>NMe_2_ [2-*N*,*N*-Dimethylamino-8-bromo-4-(4-chlorophenyl)-4*H*-1,3-dithiol[4,5-*c*]chromen-2-ylium tetrafluoroborate (**7a**)], and >piperidine [2-(Piperidin-1-yl)-8-bromo-4-(4-chlorophenyl)-4*H*-1,3-dithiol[4,5-*c*]chromen-2-ylium tetrafluoroborate (**7c**)]. It can be seen that **7e**–**l** that contained NEt_2_ at R_2_ exhibited better bacterial inhibition compared to commercialized antibiotics, kanamycin, and ampicillin. Next, halogen substitution at C6 of ring A and C4′ of ring B seemed to be important for activity since hydrogen atom substitution at those positions reduced the antibacterial activity, as can be seen in compound **7m** [89].

Farooq et al. (2020) synthesized three chalcone derivatives and evaluated their antibacterial action against *S. aureus* via disc diffusion assay and turbidimetric kinetics. The Table 14 shows that all three compounds exhibited good antibacterial properties, especially 4-(€-3-(4-Fluorophenyl)-3-oxoprop-1-enyl)benzoic acid (**8b**). It can be seen that **8b** exerted the maximum zone inhibition and the lowest MIC value compared to other synthesized compounds. **8b** was also outperformed the commercially available antibiotic ampicillin in terms of zone inhibition. The summary of SAR analysis is illustrated in Figure 24. The authors discussed that good antibacterial activity exerted by these compounds was due to the presence of COOH moiety at C2′ of ring B [90]. Then, as previously stated, **8b** demonstrated the highest activity, attributed to the fluorine substituent at C6 of ring A [91]. The authors discussed that higher electronegative substitution improved the antibacterial action [92].

Narwal et al. (2021) synthesized a variety of substituted chalcones and subjected them to antibacterial assay against several bacterial strains, including *S. aureus*, *B. subtilis*, *E. coli*, *P. aeruginosa* and *Salmonella enterica.* The antibacterial activity can be observed in Table 15. The authors analyzed several significant characteristics of synthesized chalcones for their antibacterial action as illustrated in Figure 25. To begin, substitution at C6 of ring A, such as NO_2_, NH_2_ and OH appeared to play an essential role in chalcones’ antibacterial action [93]. Other than that, electron donating groups such as hydroxyl and methoxy groups at ring B helped to boost the antibacterial activity against *B. subtilis* and *P. aeruginosa*, as observed in compounds (*E*)-3-(4-Hydroxyphenyl)-1-(4–Nitrophenyl)prop–2–en–1–one (**9a**), (*E*)–3–(3,4,5–Trimethoxyphenyl)–1–(4–Nitrophenyl)Prop–2–en–1–one (**9c**), and (*E*)–3–(4–Hydroxy–3–methoxyphenyl)–1–(4–Nitrophenyl)prop–2–en–1–one (**9f**) (MIC ≥ 1.16 μg/mL). Next, presence of bromine moiety at C2′ improved the antibacterial activity against *E. coli*, *B. subtilis* and *P. aeruginosa*, as depicted in compound (*E*)–3–(4–Bromophenyl)–1–(4–Nitrophenyl)prop–2–en–1–one (**9h**) (MIC ≥ 0.94 μg/mL) [94].

Sadgrove et al. (2020) isolated several isoflavones and derivatives from the bark of *Erythrina lysistemon*. The compounds were identified as erybraedin A, phaseollidin, abyssinone V-4′ methyl ether, eryzerin C, alpumisoflavone, cristacarpin, and lysisteisoflavone. These compounds’ antibacterial activity was also tested, and the findings are reported in Table 16. The authors addressed how prenylation is vital for activity because all molecules except alpumisoflavone have a prenyl group. Following that, hydroxylation at the same aromatic ring bearing the prenyl group can improve antibacterial activity. As demonstrated in abyssinone V-4′ methyl ether and cristacarpin, methoxylation of those hydroxyl groups decreased activity (Figure 26) [95].

Septama et al. (2020), on the other hand, extracted two flavonoids (viz., dihydromorin and norartocarpetin) from *Artocarpus heterophyllus* and tested them against six different bacterial strains. Table 17 clearly shows that dihydromorin inhibited bacteria better than norartocarpetin against all strains tested. Therefore, based on Figure 27, it can be seen that hydroxylation of C3 of ring C and saturated C2-C3 lead to better bacterial inhibitory properties [96].

Nielsen et al. (2004) synthesized a series of carboxylic chalcones and executed an in vitro antibacterial assay against *S. aureus*. The results of the antibacterial activity can be observed in Table 18. Some important structural features of carboxylic chalcones as antibacterial agents have been discussed. Firstly, substitution in ring B at *meta* position [4′-Carboxy-3-trifluoromethyl-chalcone (**10h**), 4′-Carboxy-3-bromo-chalcone (**10i**), 4′-Carboxy-3-chloro-chalcone (**10j**), 4′-Carboxy-3-nitro-chalcone (**10k**), 4′-Carboxy-3-hydroxy-chalcone (**10l**), and 4′-Carboxy-3-phenoxy-chalcone (**10m**); (MIC ≥ 40 μM) led to a more potent inhibition compared to *para* substitution [4′-Carboxy-4-trifluoromethyl-chalcone (**10b**), 4′-Carboxy-4-chloro-chalcone (**10c**), 4′-Carboxy-4-methyl-chalcone (**10d**), 4′-Carboxy-4-methoxy-chalcone (**10e**), 4′-Carboxy-4-hydroxy-chalcone (**10f**), and 4′-Carboxy-4-phenoxy-chalcone (**10g**); (MIC ≥ 150 μM). Next, the presence of lipophilic substituents such as bromine, chlorine and CF_3_ as can be seen in **10b**,**c**, **10h**–**j**, 4′-Carboxy-2-trifluoromethyl-chalcone (**10n**), 4′-Carboxy-2-bromo-chalcone (**10o**), 4′-Carboxy-2-chloro-chalcone (**10p**), 4′-Carboxy-3,5-bis-trifluoromethyl-chalcone (**10s**), 4′-Carboxy-3,5-dibromo-chalcone (**10t**), and 4′-Carboxy-3,5-dichloro-chalcone (**10u**) (MIC ≥ 2 μM) showed better antibacterial action than those compounds possessing polar hydroxy and methoxy groups (**10e**,**f**,**l**), 4′-Carboxy-2-hydroxy-chalcone (**10q**), and 4′-Carboxy-3,5-dimethoxy-chalcone (**10x**); (MIC > 300 μM) (Figure 28). The importance of lipophilic substitution was further supported by the fact that the most active compounds (**10s**,**t**; MIC ≥ 2 μM) contained two lipophilic substituents with potencies equivalent to two different antibiotics, namely ciprofloxacin and linezolid [97].

Ansari et al. (2009) developed a series of chalcones and evaluated their antibacterial activity against Gram-negative bacteria viz., *Bordetella bronchiseptica*. The bioactivity results are shown in Table 19. Results of the study were found similar to the previous findings reported by Nielsen et al. (2004), lipophilic substitution at the different positions in ring B brought to a better chalcones’ antibacterial activity (Figure 29). It can be seen that chalcone derivatives with lipophilic substituents such as 1-(3′-Hydroxyphenyl)-3-(2-chlorophenyl)-2-propen-1-one (**11m**), 1-(3′-Hydroxyphenyl)-3-(3-chlorophenyl)-2-propen-1-one (**11n**), 1-(3′-Hydroxyphenyl)-3-(4-chlorophenyl)-2-propen-1-one (**11o**) and 1-(3′-Hydroxyphenyl)-2-(3-bromophenyl)-2-propen-1-one (**11p**) with MIC value of ≥0.2 mg/mL showed better antibacterial action than those derivatives which had polar hydroxy and methoxy groups such as 1-(3′-Hydroxyphenyl)-3-(2-hydroxyphenyl)-2-propen-1-one (**11b**), 1-(3′-Hydroxyphenyl)-3-(3-hydroxyphenyl)-2-propen-1-one (**11c**), 1-(3′-Hydroxyphenyl)-3-(4-hydroxyphenyl)-2-propen-1-one (**11d**), 1-(3′-Hydroxyphenyl)-3-(2-methoxyphenyl)-2-propen-1-one (**11e**), 1-(3′-Hydroxyphenyl)-3-(3-methoxyphenyl)-2-propen-1-one (**11f**), 1-(3′-Hydroxyphenyl)-3-(4-methoxyphenyl)-2-propen-1-one (**11g**), 1-(3′-Hydroxyphenyl)-3-(3,4-dimethoxyphenyl)-2-propen-1-one (**11h**), 1-(3′-Hydroxyphenyl)-3-(4-hydroxy,3-methoxyphenyl)-2-propen-1-one (**11i**), and 1-(3′-Hydroxyphenyl)-3-(2-methyl-3,5-dimethoxyphenyl)-2-propen-1-one (**11w**) (MIC > 0.6 mg/mL). Furthermore, some polar compounds, such as **11h**,**i**,**w**, displayed no activity [98].

Next, Woznicka et al. (2013) investigated the antibacterial activities of flavonoids and their sulfonic derivatives. *E. coli*, *P. aeruginosa*, and *S. aureus* were among the bacterial strains used in the study. Meanwhile, the tested compounds include quercetin, morin, sodium salt of quercetin-5′-sulfonic acid (NaQSA), and sodium salt of morin-5′-sulfonic acid (NaMSA). The bacterial action of the tested compounds can be viewed in Table 20. To start, it was discussed that hydroxylation at C5 and C7 of ring A is important for antibacterial activity because all compounds contained it [68]. Next, it can be seen that morin with 2′,4′-OH substitution exhibited better activity than the quercetin with 3′,4′-OH substitution, which proved the importance of hydroxylation at C2′ of ring B in bacterial inhibition [59,99]. Next, further investigation revealed that the presence of a sulfo group at C5’ was only found to enhance bacterial activity against *S. aureus* and not against any of the other tested strains. The summary of SAR for flavonoids and their sulfonic derivatives as antibacterial agents is presented in Figure 30 [100].

Oliveira et al. (2021) investigated the effects of fatty acid chain length differences in anthocyanins, particularly cyanidin-3-glucoside, on antibacterial activity. The glucoside with a fatty acid chain was inserted at C3 of cyanidin’s ring A, and the length of the fatty acid chain ranged from C4 to C12. Their bacterial inhibition can be observed in Table 21. Compared to other compounds with fatty acid chains, cyanidin-3-*O*-glucoside (**12a**) with no fatty acid chain was found inactive. Following that, it was discovered that the chain length comprising 12 carbons in the sugar moiety [cyanidin-3-*O*-glucoside-C12 (**12f**)] was found too long to exert antibacterial action, whereas the chain length containing 4 carbons [cyanidin-3-*O*-glucoside-C4 (**12b**)] was too short. It can be concluded that the optimal size for antibacterial activity was six carbons to ten carbons, where cyanidin-3-*O*-glucoside-C10 (**12e**) with a fatty acid chain comprising ten carbons was found out to be the most potent compound. The important structural features of cyanidin as antibacterial agent can be viewed in Figure 31 [101].

Shoaib et al. (2020) synthesized three halogenated flavone derivatives and tested their antibacterial activity against *B. subtitlis*, *S. aureus*, and *P. aeruginosa*. According to Table 22, 6-bromo-2-(4-(trifluoromethyl)phenyl)-4H-chromen-4-one (**13c**) was the most active compound against all of the tested strains. Based on the MIC value of **13c**, it can be concluded that **13c** was antibacterially equipotent to ciprofloxacin, particularly against *S. aureus* and *P. aeruginosa*. As a result, based on the efficacy of **13c**, it can be concluded that halogenation at C6 of ring A enhanced the antibacterial action. The authors then discussed how the trifluoromethyl group at C4′ of ring B appeared to be important for bacterial inhibition because all tested compounds contained this moiety and exerted antibacterial properties. The SAR analysis of tested flavones is illustrated in Figure 32 [102].

Sato et al. (2006) tested the antibacterial activity of some isoflavones, namely isolupalbigenin, erythrinin B, lupiwighteone, and laburnetin against eleven strains of MRSA. The potency of these compounds was found to be in the given order, isolupalbigenin (MIC_50_ = 3.13 μg/mL) > erythrinin B (MIC_50_ = 6.25 μg/mL) > lupiwighteone = laburnetin (MIC_50_ > 25 μg/mL). Results revealed that the prenyl group at C3′ of ring B as presented in isolupalbigenin led to increased MRSA inhibition. Similarly, as seen in erythrin B, prenylation at C6 of the ring improved anti-MRSA activity. Following that, all of the tested compounds had hydroxyl groups at C5 of ring A, indicating that hydroxylation at C5 is critical for antibacterial activity against MRSA. The summarization of the SAR can be viewed in Figure 33 [103].

Fang et al. (2016) used 3D-QSAR and docking studies to investigate the SAR of flavonoids as potential *E. coli* inhibitors. The study was conducted against DNA gyrase B (GyrB) of *E. coli*. The findings from both 3D-QSAR and docking investigations revealed that hydroxylation at the C3, C5, and C7 positions of ring A was critical in inhibiting *E. coli*. It was also discovered that the carbonyl group at position C4 plays a vital function in *E. coli* inhibition. Following that, 3D-QSAR demonstrated that hydroxylation at C3′ of ring B, as well as the presence of C2=C3, were both required for GyrB inhibition. On the other hand, C6 hydroxylation and C8, C3′ methoxylation can all result in a decrease in *E. coli* inhibition. The summary of SAR discussion is depicted in Figure 34 [104].

Alfonso et al. (2018) synthesized six A-type procyanidin analogues (viz., 2-(3′,4′-Dihydroxyphenyl)chromane-(4 → 8,2 → O-7)-catechin (**1**); 2-(3′,4′-Dihydroxyphenyl)-5,7-dihydroxychromane-(4 → 4″,2 → O-5″)-phloroglucinol (**2**); 2-(3′,4′-Dihydroxyphenyl)-7-hydroxychromane-(4 → 4″,2 → O-5″)-phloroglucinol (**3**); 2-(3′,4′-sDihydroxyphenyl)-6-nitrochromane-(4 → 4″,2 → O-5″)-phloroglucinol (**4**); 3-Chloro-2-(3′,4′-dihydroxyphenyl)chromane-(4 → 4″,2 → O-5″)-phloroglucinol (**5**); 3-Chloro-2-(3′,4′-diacetoxyphenyl)chromane-(4 → 4″,2 → O-5″)-1,3-diacetylphloroglucinol (**6**)); evaluated their antimicrobial and antibiofilm properties against resistant bacteria, isolated from organic foods (12 Gram-positive and Gram-negative bacteria). This research work was carried out to find the structural features responsible for their antimicrobial and antibiofilm activities. Among all the synthesized compounds, 2-(3′,4′-sDihydroxyphenyl)-6-nitrochromane-(4 → 4″,2 → O-5″)-phloroglucinol, which had a NO_2_ group on ring A, exhibited the highest antibacterial activity (MIC of 10 μg/mL). The same compound also displayed the best effect at preventing biofilm formation (up to 40% decreases at 100 μg/mL) and disrupting preformed biofilms (up to 40% reductions at 0.1 μg/mL). As depicted in Figure 35, structure–activity relationships findings showed that electron withdrawing group attached to ring A was found to be responsible to manifest strong antibacterial action [105].

Moreover, another similar study conducted by Ortega-Vidal et al. (2022) in which nine phenolic compounds (viz. 2 flavan-3-ols (catechin and epicatechin); 1 phenolic glucoside (annphenone); 6 dimeric A-type proanthocyanidins) were isolated from the pruning wood residue of European plum (*Prunus domestica* L.). All the isolated compounds were further subjected to determine their antibacterial and antibiofilm effects against a selection of multi-resistant strains including foodborne microorganisms. The most potent antibacterial activity was observed on the strain *Enterobacter* sp. UJA37p (obtained from organic tomato) and with high tolerance to different biocides. All phenolic compounds exhibited MIC values of 100 μg/mL against this resistant bacterium. In contrast, when bacteria were previously allowed to attach and form biofilm before treatments, the interesting results were observed in which disruption of preformed biofilms was noticed with (+)-epiafzelechin-(2β → *O*→7,4β → 8)-epicatechin, followed by (−)-*ent*-epiafzelechin-(2α → *O* → 7,4α → 8)-catechin and i.e., annphenone at a concentration of 0.1 μg/mL further confirming the potent antibacterial action of these phenolic compounds. The chemical structures of mentioned flavonoids can be viewed in Figure 36 [106].

The extensive discussion of flavonoids possessing antibacterial properties and their structural requirements for antibacterial activity led us to conclude numerous structural requirements frequently noted in the reviewed literature. Table 23 summarizes the majority of the SAR discussed before for several classes of flavonoids, including chalcones, flavanones, flavones, isoflavones, flavan-3-ols, and anthocyanidins. Significant structural traits can only be compared and concluded for chalcones, flavanones, flavones, and isoflavones, as flavan-3-ols and anthocyanidins lack sufficient comparison.

Table 23 shows that hydroxylation at C5 and C7, as well as lipophilic substitution at ring B improves chalcones’ antibacterial activity. Following that, several authors have observed that C5 hydroxyl group and C2-C3 double bond saturation can enhance the antibacterial activity of flavanones. Hydroxylation of flavanones at ring B can enhance the antibacterial activity, while methylation or glycosylation of the hydroxyls can reduce their antibacterial activity. Next, hydroxylation at C5 can enhance flavones’ antibacterial activity, whilst methoxylation at C5 can decrease it. C7-OH, C4′-OH, C3-OH, and C2=C3 double bond have all been observed to enhance the antibacterial activity of flavones. In the case of isoflavones, hydroxylation at rings A and B, particularly at C5 and C4′, as well as prenylation at ring A, particularly at C6, might enhance antibacterial action. On the other hand, methoxylation and glycosylation in ring A and B seemed to reduce the antibacterial activity.

Then, it can be seen that there is some inconsistency in the findings, specifically whether hydroxylation and methylation of the hydroxyl groups boosted or lowered the antibacterial activity of flavonoids, specifically at the C6 and C3′ positions of flavones. This inconsistency can be explained by the significance of antibacterial agents’ amphiphilic balance to enter bacterial strains, where both hydrophilic and hydrophobic characteristics must be present in balance [76].

According to the summary above, modifications to each flavonoids class, particularly in synthetic or semi-synthetic approaches, may be made to enhance the antibacterial activity. However, when looking at the broader picture without regard to the flavonoids class, it was shown that hydroxylation of C5, C7, C3′, and C4′ as well as geranylation or prenylation at C6 are important structural requirements for antibacterial activity. On the other hand, it has been shown that methoxylation at C3′ and C5 reduces the antibacterial activity of flavonoids. Yet, it should be highlighted that flavonoids’ local balance of amphiphilicity will still be required for bacterial penetration to exert potent antibacterial effects. Moreover, after thorough review of all the antibacterial research studies of flavonoids, it is believed that the above-mentioned key structural features may play a critical role to orchestrate strategies in the modern chemical synthesis to discover more potent flavonoids’ based antibacterial drugs.

## 4. Conclusions

Flavonoid-protein interactions (enzymes, receptors, transporters, and transcription factors) are basic phenomena that govern the beneficial effects of flavonoids. As has been mentioned in the review, one of the most studied pharmacological effects of flavonoids is their antibacterial property which has been thoroughly evaluated through various SAR studies to discover more potent antibacterial agents as safe natural products. It can be concluded that several flavonoids’ structural features may be important for imparting antibacterial effects, including C5, C7, C3′, and C4′ hydroxylation as well as geranylation or prenylation at C6. However, the most important aspect of flavonoids is that they must retain their amphiphilic characteristics to penetrate bacteria to exert their potent antibacterial action. Hence, these important structural features of antibacterial flavonoids, if taken into account while orchestrating new synthetic strategies, might play an important role to synthesize better antibacterial drugs to overcome stiff challenges associated with the resistant bacteria.

## Figures and Tables

**Figure 1 molecules-27-01149-f001:**
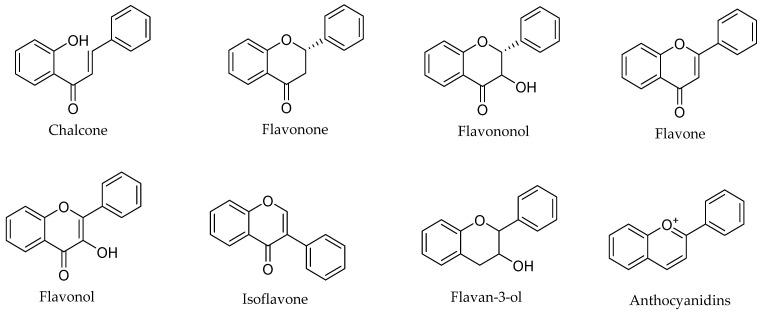
Various classes of flavonoids.

**Figure 2 molecules-27-01149-f002:**
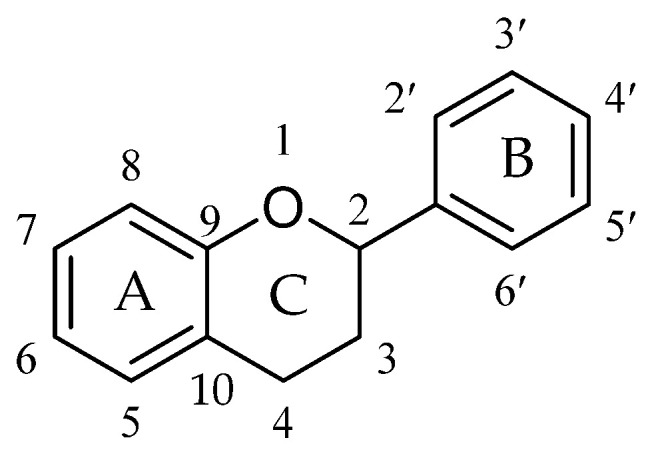
Basic skeleton and numbering system of flavonoids.

**Figure 3 molecules-27-01149-f003:**
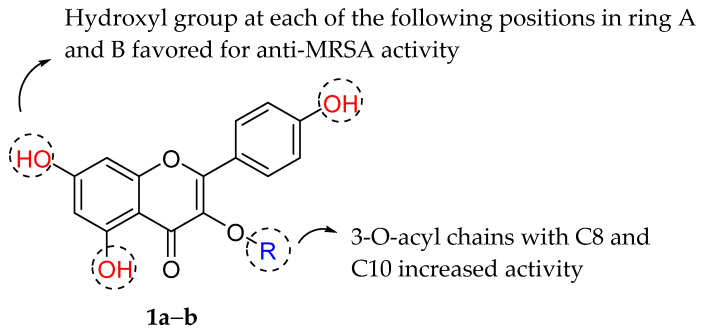
SAR analysis of tested flavonoids [70].

**Figure 4 molecules-27-01149-f004:**
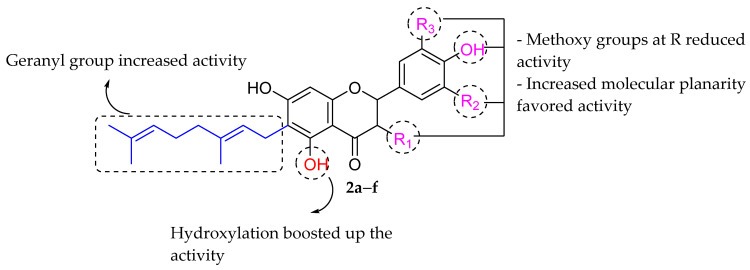
SAR analysis of isolated flavonoids from *Paulownia tomentosa* [71].

**Figure 5 molecules-27-01149-f005:**
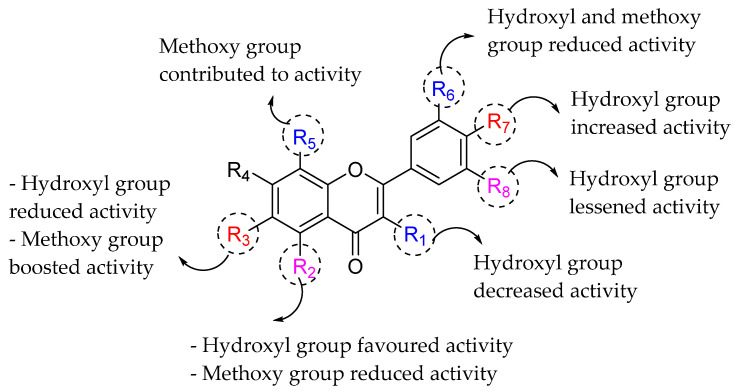
The SAR of flavonoids as antibacterial agents by inhibiting the DNA gyrase activity [62].

**Figure 6 molecules-27-01149-f006:**
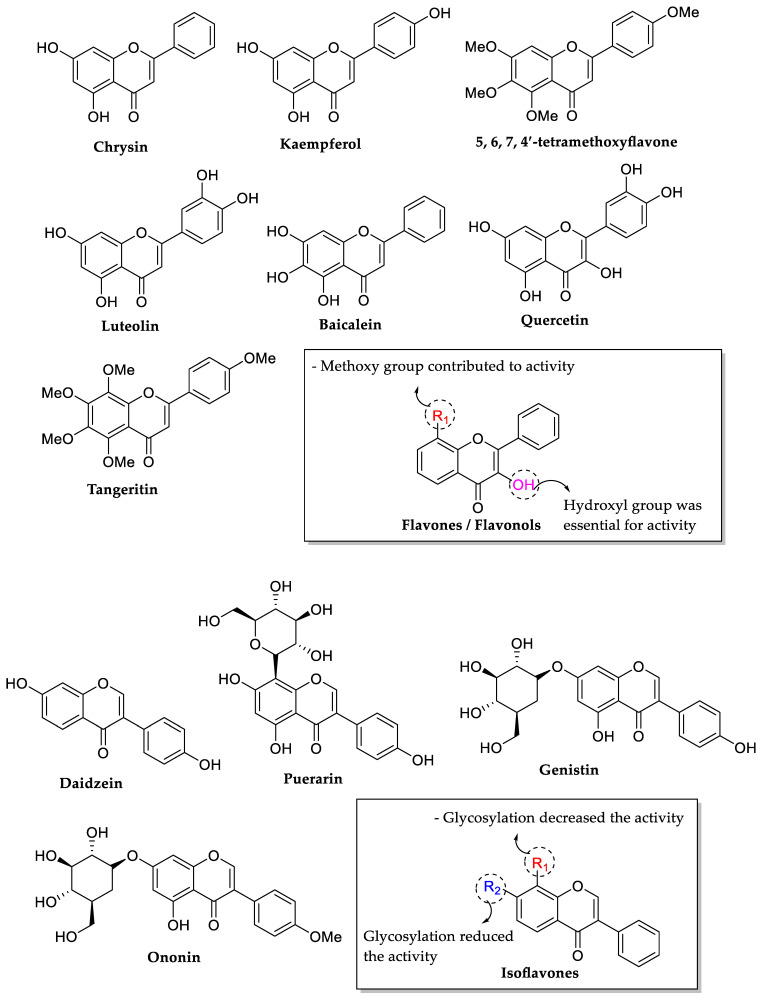
The SAR of flavonoids as antibacterial by inhibiting the growth of *E. coli* [73].

**Figure 7 molecules-27-01149-f007:**
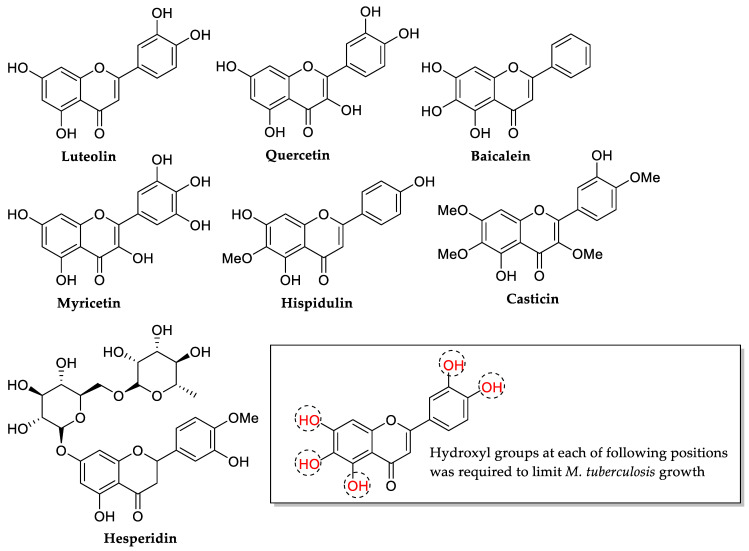
Chemical structures of several tested flavonoids and their specific structural requirements for antitubercular activity (*Mycobacterium tuberculosis*) [74].

**Figure 8 molecules-27-01149-f008:**
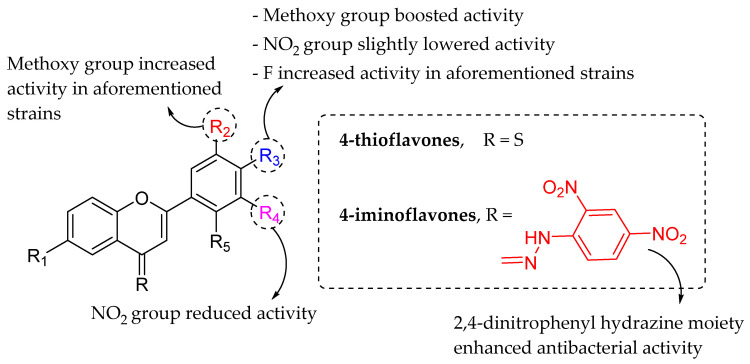
The structural-activity analysis of 4-thioflavones and 4-iminoflavones [75].

**Figure 9 molecules-27-01149-f009:**
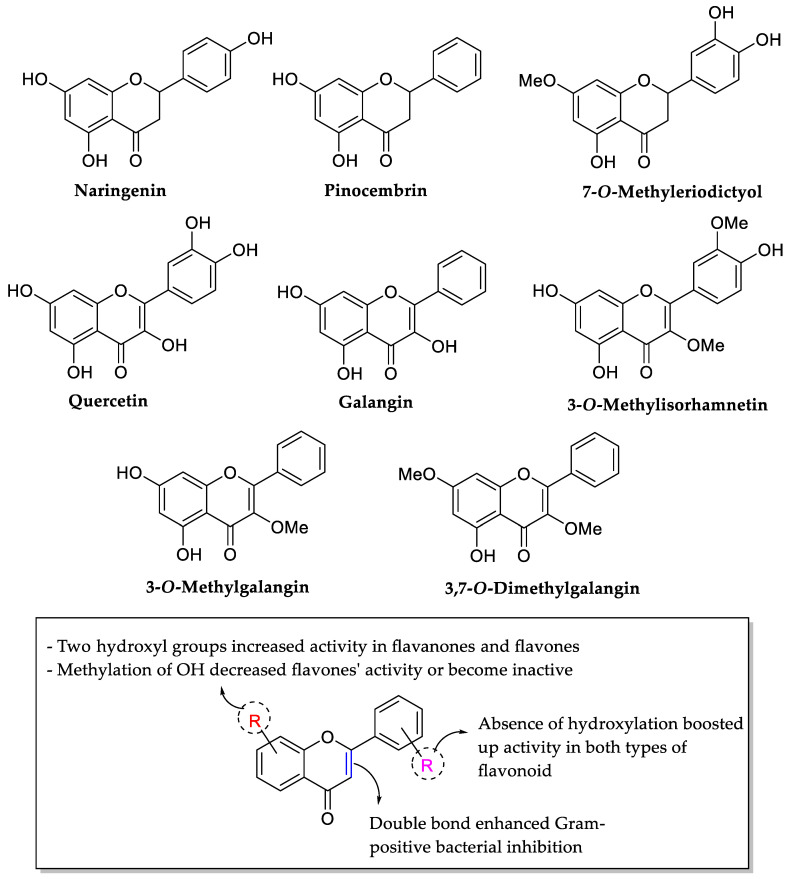
Chemical structure of several tested flavonoids and the SAR of flavonoids studied for antibacterial properties against four Gram-positive and four Gram-negative bacteria [76].

**Figure 10 molecules-27-01149-f010:**
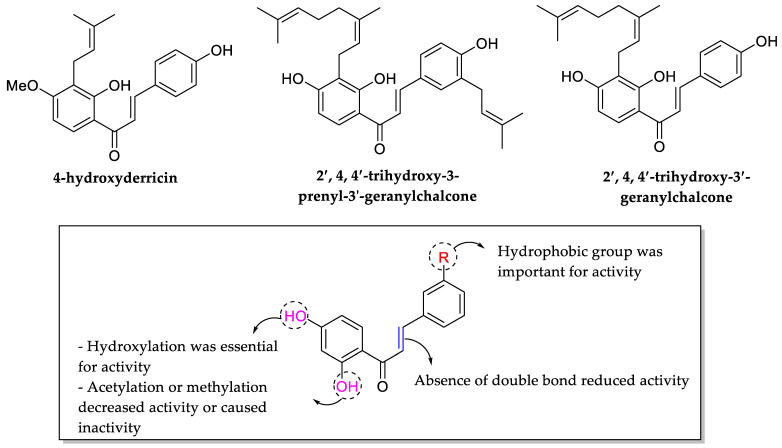
Chemical structures of several tested chalcones and SAR of chalcones for antibacterial activity [77].

**Figure 11 molecules-27-01149-f011:**
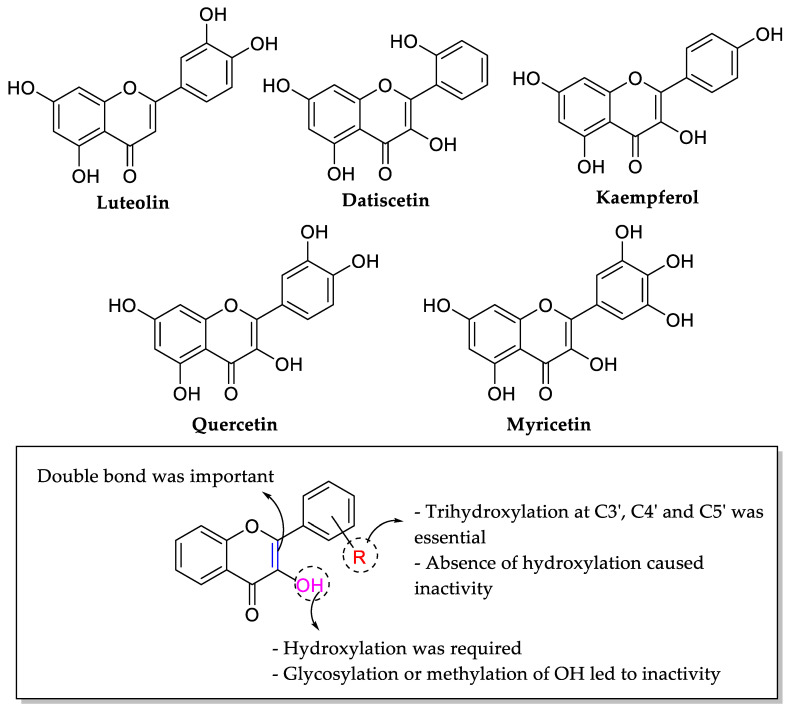
Chemical structures of tested flavonoids that were active and the structural feature requirements for the manifestation of flavonoids’ antibacterial activity against antibiotic resistant bacteria [64].

**Figure 12 molecules-27-01149-f012:**
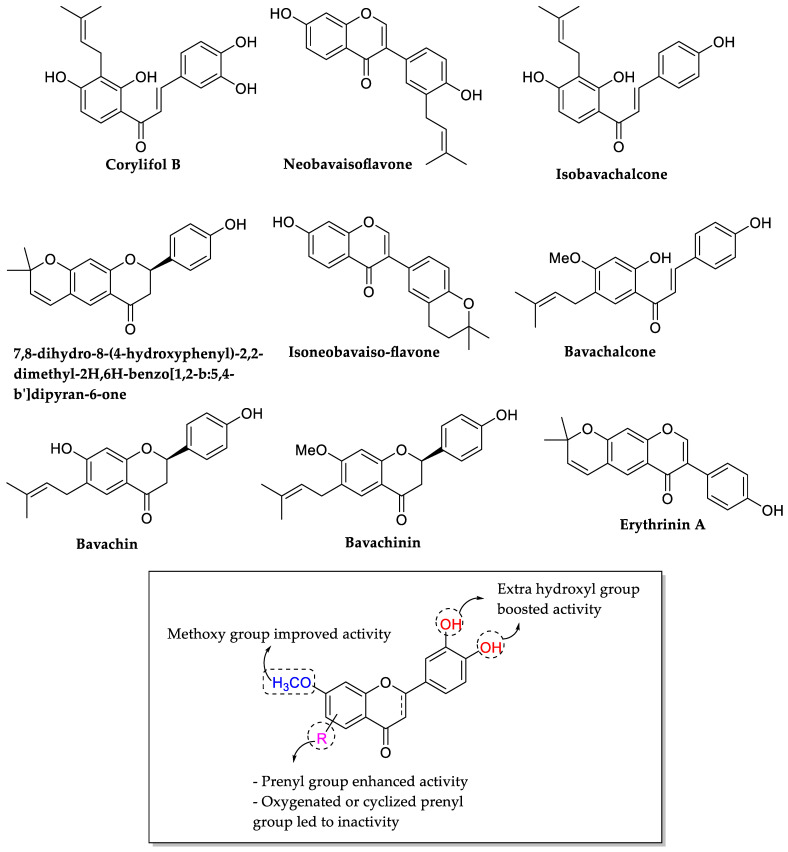
Chemical structures of isolated flavonoids that have excellent inhibition and the summary of important structural features of flavonoids as antibacterial agents isolated from *Psoralea corylifolia* seeds [78].

**Figure 13 molecules-27-01149-f013:**
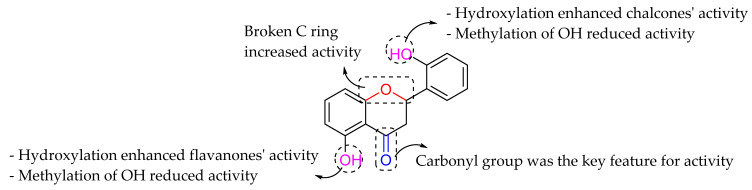
Important SAR of flavonoids as antibacterial agent against MRSA [79].

**Figure 14 molecules-27-01149-f014:**
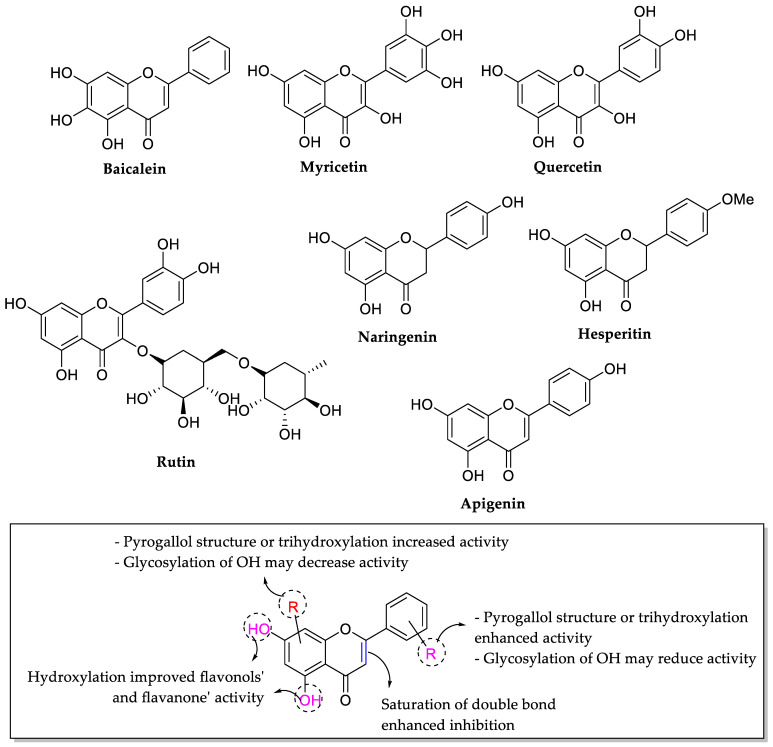
Chemical structures of mentioned flavonoids and their related SAR for antibacterial activity [80].

**Figure 15 molecules-27-01149-f015:**
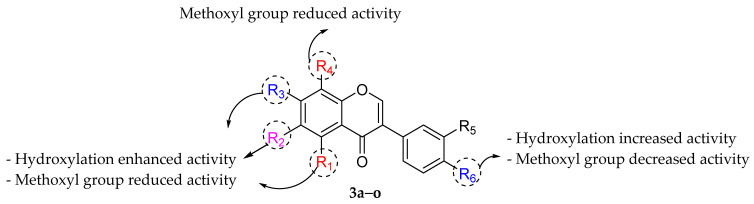
SAR of isoflavones as antibacterial agent [81].

**Figure 16 molecules-27-01149-f016:**
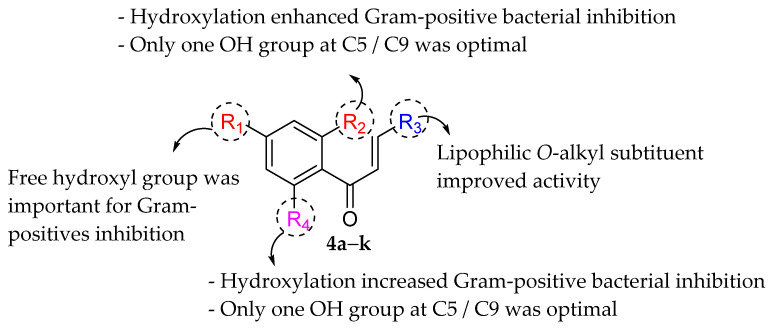
The chemical structure and SAR of chalcones tested [82].

**Figure 17 molecules-27-01149-f017:**
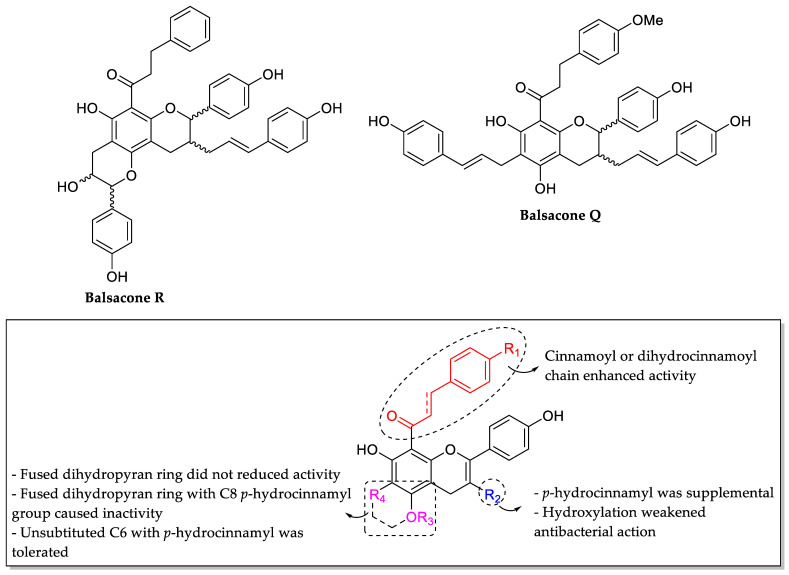
Chemical structures of Balsacone R and Balsacone Q, with the SAR of balsacones for their anti-MRSA activity [83].

**Figure 18 molecules-27-01149-f018:**
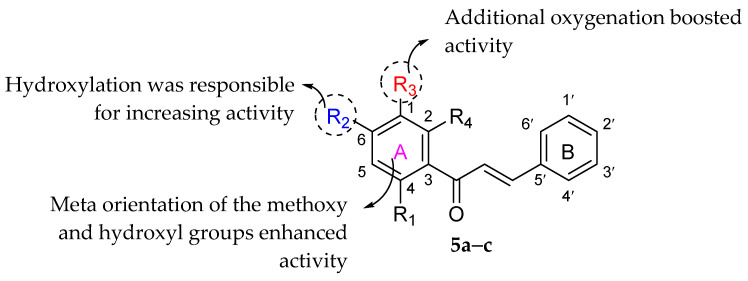
Structural activity of chalcones against MRSA strains [84].

**Figure 19 molecules-27-01149-f019:**
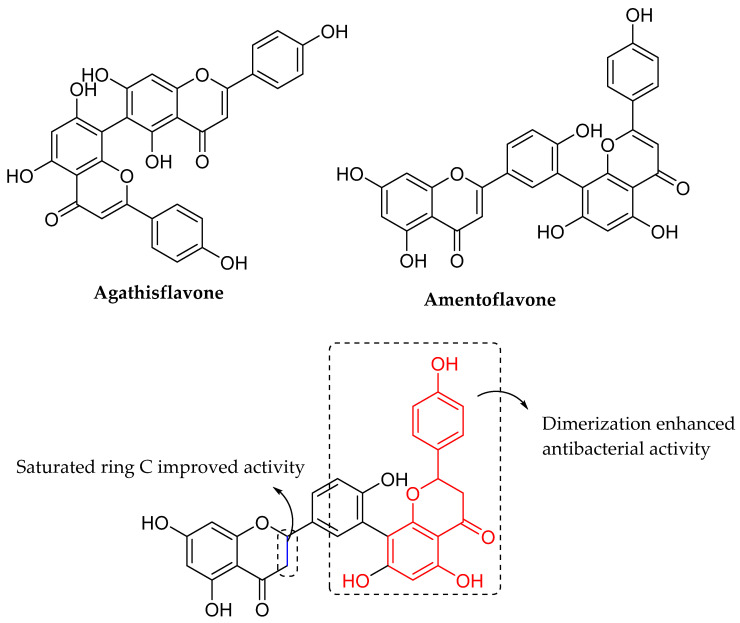
The chemical structures of biflavonoids isolated from *Schinus terebinthifolius* Raddi fruits [85].

**Figure 20 molecules-27-01149-f020:**
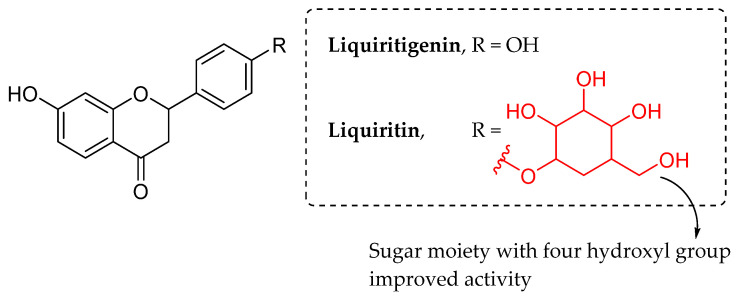
The chemical structure of tested flavonoids [86].

**Figure 21 molecules-27-01149-f021:**
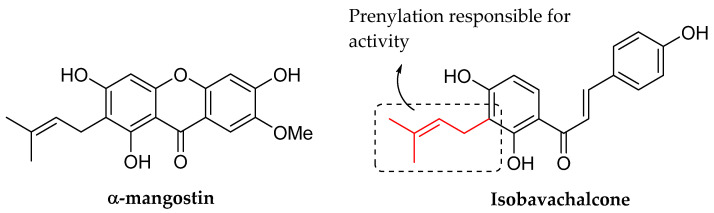
α-mangostin, isobavachalcone and structural feature responsible for antibacterial activity [65].

**Figure 22 molecules-27-01149-f022:**
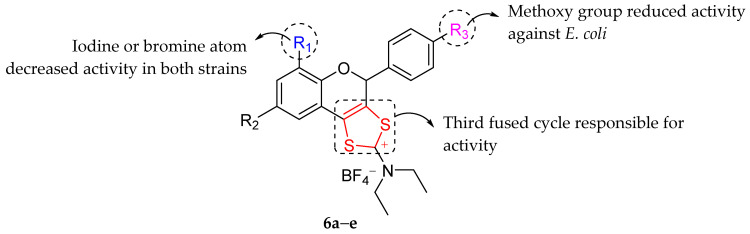
Summary of SAR of sulfur-containing tricyclic flavonoids against *S. aureus* and *E. coli* [88].

**Figure 23 molecules-27-01149-f023:**
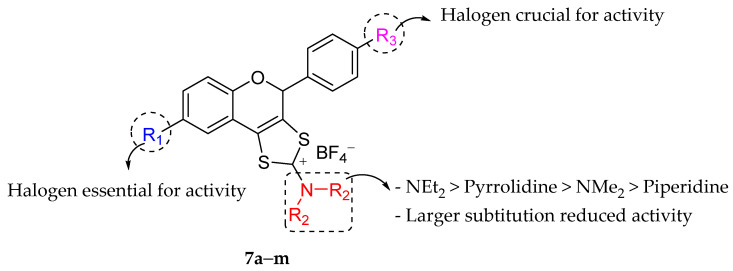
SAR analysis of sulfur-containing tricyclic flavonoids against *S. aureus* and *E. coli* [89].

**Figure 24 molecules-27-01149-f024:**
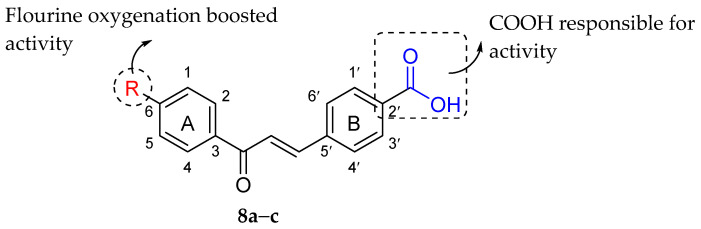
SAR of chalcones against *S. aureus* [92].

**Figure 25 molecules-27-01149-f025:**
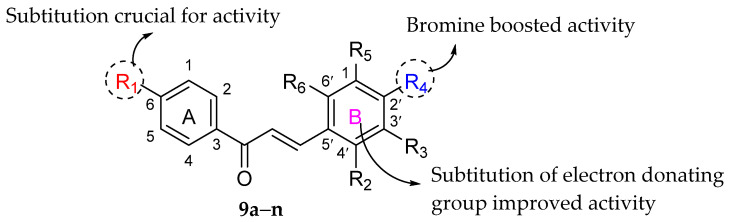
Important structural features of chalcones’ antibacterial activities [94].

**Figure 26 molecules-27-01149-f026:**
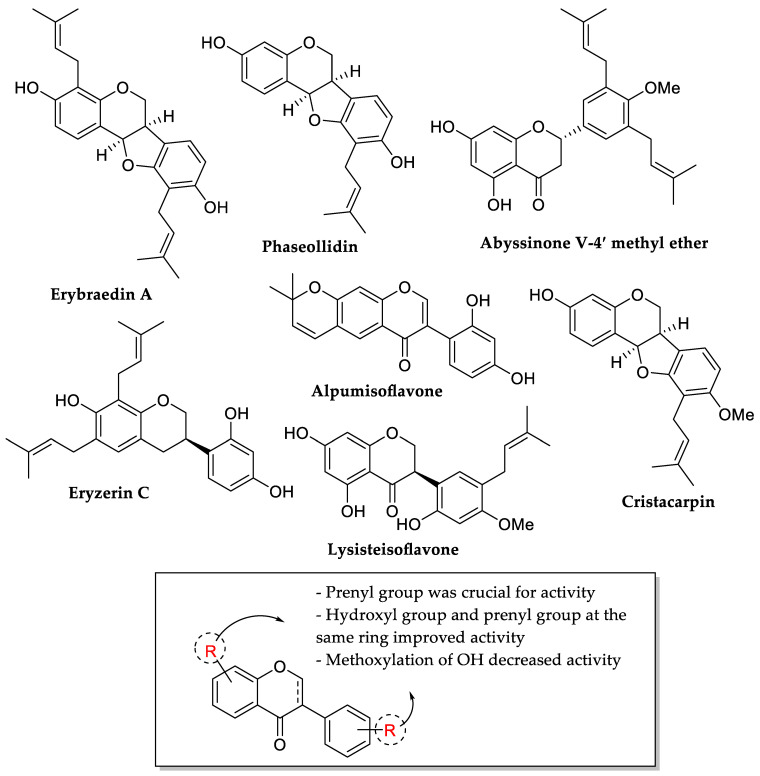
Structures of isoflavones and derivatives from the bark of *Erythrina lysistemon* and the SAR analysis of isoflavones as antibacterial agents [95].

**Figure 27 molecules-27-01149-f027:**
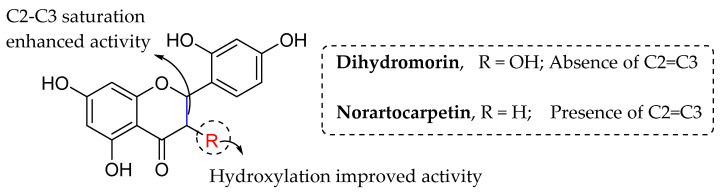
SAR of flavonoids isolated from *Artocarpus heterophyllus* as antibacterial agents [96].

**Figure 28 molecules-27-01149-f028:**
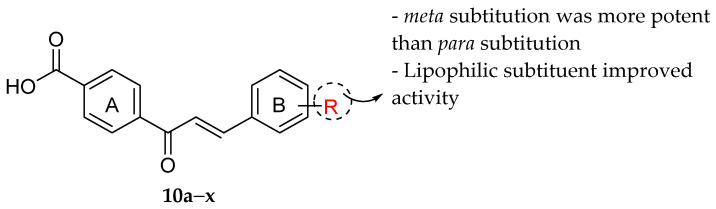
SAR of carboxylated chalcones for antibacterial activities [97].

**Figure 29 molecules-27-01149-f029:**
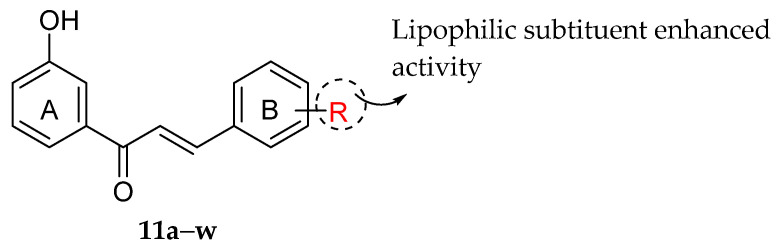
Important feature for chalcone containing non-polar groups to exhibit antibacterial activity against *B. bronchiseptica* [98].

**Figure 30 molecules-27-01149-f030:**
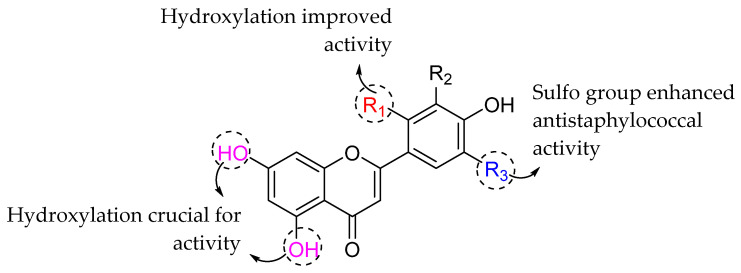
SAR analysis of flavonoids and their sulfonic derivatives as antibacterial agents [100].

**Figure 31 molecules-27-01149-f031:**
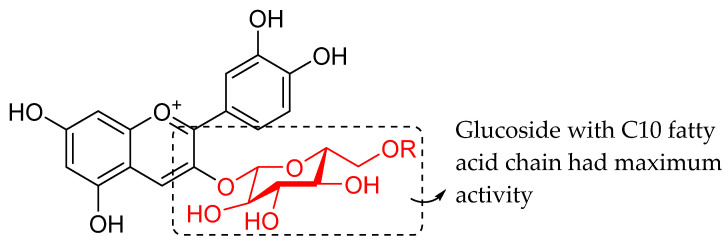
Key structural feature of cyanidin as antibacterial agent [101].

**Figure 32 molecules-27-01149-f032:**
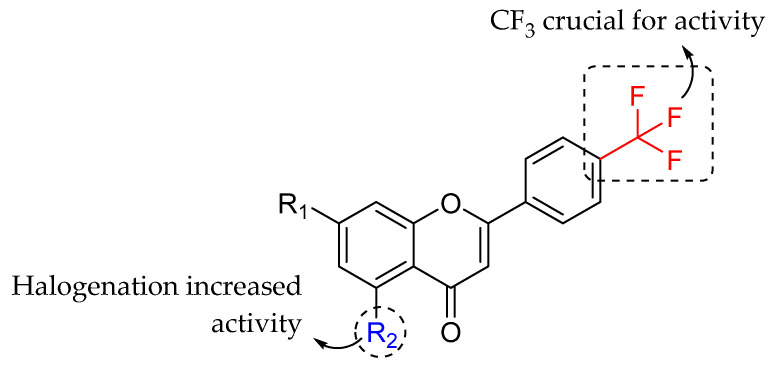
The structural-activity analysis of tested flavone derivatives containing halogens [102].

**Figure 33 molecules-27-01149-f033:**
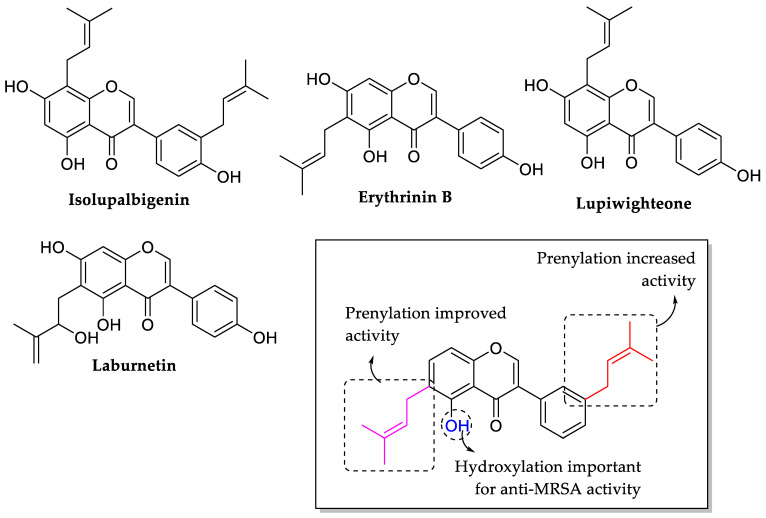
The chemical structures and the SAR of tested isoflavones [103].

**Figure 34 molecules-27-01149-f034:**
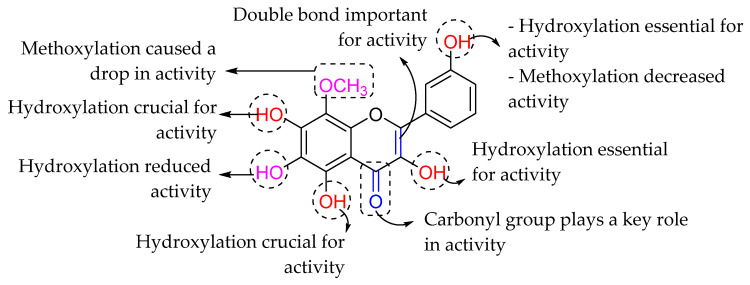
Summary from the 3D-QSAR and docking studies [104].

**Figure 35 molecules-27-01149-f035:**
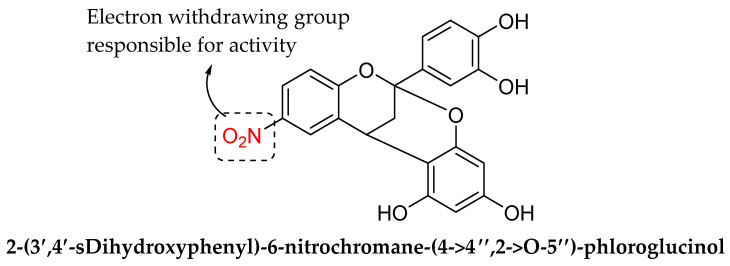
The chemical structure of 2-(3′,4′-sDihydroxyphenyl)-6-nitrochromane-(4 → 4″,2 → O-5″)-phloroglucinol [105].

**Figure 36 molecules-27-01149-f036:**
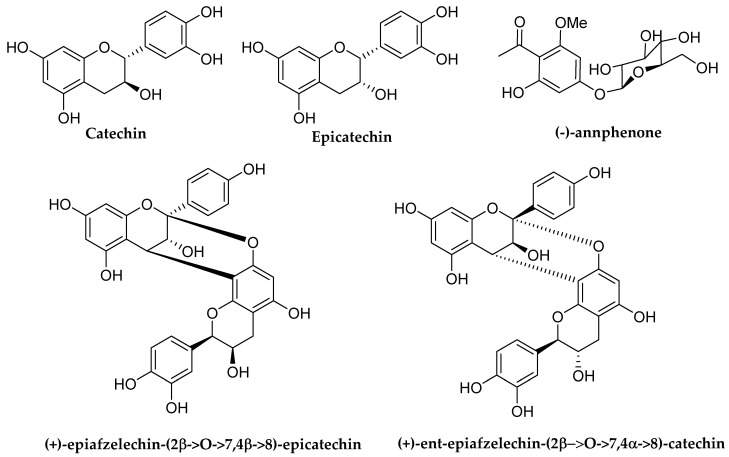
The chemical structures of mentioned flavonoids [106].

**Table 1 molecules-27-01149-t001:** Isolated compounds from *Laurus nobilis* were shown to demonstrate potent antibacterial activity against MRSA strains.

Com. No.	R	Anti-MRSA Activity (MIC: μg/mL)						
		MRSA Strains						
		OM481	OM505	OM584	OM623	* COL	N315	209P
*** 1a**	α-L-(2‴,4‴-di-*E*-*p*-coumaroyl)-rhamnoside	1	1	2	2	1	1	0.5
*** 1b**	α-L-(2″-*Z*-*p*-coumaroyl-4‴-*E*-*p*-coumaroyl)-rhamnoside	1	2	2	2	1	1	0.5
Oxacillin	-	512	128	256	256	512	8	0.13
Ciprofloxacin	-	8	1	16	8	<0.13	0.25	<0.13
Norfloxacin	-	128	8	64	64	1	2	0.5
Erythromycin	-	>1024	>128	>1023	>128	>0.12	>1024	2
Tetracycline	-	4	0.25	128	64	128	0.13	0.13

* Kaempferol 3-*O*-α-l-(2‴,4‴-di-*E*-*p*-coumaroyl)-rhamnoside (**1a**) and Kaempferol 3-*O*-α-l-(2″-*Z*-*p*-coumaroyl-4‴-*E-p*-coumaroyl)-rhamnoside (**1b**); * COL, *Colindale*.

**Table 2 molecules-27-01149-t002:** Isolated compounds from *Paulownia tomentosa* were showed to display potent antibacterial activity against several Gram-positive strains.

Com. No.	R_1_	R_2_	R_3_	MIC (μg/mL)					
				* B.c	* B.s	* E.f	* L.m	* S.a	* S.e
*** 2a**	H	OMe	OH	4	4	4	4	2	4
*** 2b**	H	OMe	OMe	4	4	4	4	4	4
*** 2c**	OH	OMe	H	2	4	4	2	2	2
*** 2d**	H	OMe	H	4	8	8	4	8	4
*** 2e**	H	H	H	4	4	4	4	8	4
*** 2f**	H	OH	H	4	4	4	4	4	4
Ciprofloxacin and nystatin	-	-	-	1	2	1	1	0.5	1

* 3′-*O*-methyl-5′-hydroxydiplacone (**2a**), 3′-*O*-methyl-5′-*O*-methyldiplacone (**2b**), 3′-*O*-Methyldiplacol (**2c**), 3′-*O*-Methyldiplacone (**2d**), mimulone (**2e**), and diplacone (**2f**); * B.c., *Bacillus cereus*; B.s., *Bacillus subtilis*; E.f., *Enterococcus faecalis*; L.m., *Listeria monocytogenes*; S.a., *S. aureus*; S.e., *Staphylococcus epidermidis*.

**Table 3 molecules-27-01149-t003:** Flavonoids compounds displayed good *E. coli* DNA gyrase inhibition.

Flavonoids	R_1_	R_2_	R_3_	R_4_	R_5_	R_6_	R_7_	R_8_	MIC_50_ (μg/mL)
Tangeritin	H	OMe	OMe	OMe	OMe	H	OMe	H	137
5,6,7,4′-tetramethoxyflavone	H	OMe	OMe	OMe	H	H	OMe	H	156
Nobiletin	H	OMe	OMe	OMe	OMe	OMe	OMe	H	177
Chrysin	H	OH	H	OH	H	H	H	H	37
Galangin	OH	OH	H	OH	H	H	H	H	53
Quercetin	OH	OH	H	OH	H	OH	OH	H	36
Baicalein	H	OH	OH	OH	H	H	H	H	71
Luteolin	H	OH	H	OH	H	OH	OH	H	67
Kaempferol	OH	OH	H	OH	H	H	OH	H	25
Myricetin	OH	OH	H	OH	H	OH	OH	OH	142

**Table 4 molecules-27-01149-t004:** Antibacterial activity of several flavones and isoflavones against *E. coli* via membrane interaction effect.

Flavones/Flavonols	MIC_50_ (μg/mL)	Isoflavones	MIC_50_ (μg/mL)
Chrysin	36.72	Daidzein	120.0
Kaempferol	25.00	Puerarin	1500
5,6,7,4′-tetramethoxyflavone	156.3	Genistin	238.0
Luteolin	67.25	Ononin	712.5
Baicalein	70.94		
Quercetin (Flavonol)	35.76		
Tangeritin	137.1		

**Table 5 molecules-27-01149-t005:** Antibacterial activity of flavanones and flavones against Gram-positive and Gram-negative bacteria.

Flavonoids	MIC (μg/μL)							
	*E. cloacae*	*E. coli*	*K. pneumoniae*	*P. mirabilis*	*B. cereus*	*B. coagulans*	*B. subtilis*	*S. aureus*
Naringenin	2	4	>4	2	2	2	2	>4
Pinocembrin	1	4	1	4	2	1	1	>4
7-*O*-Methyleriodictyol	2	4	>4	0.5	2	1	2	4
Quercetin	>4	>4	>4	0.5	2	2	>4	2
Galangin	1	1	0.5	0.25	0.25	0.25	0.25	0.5
3-*O*-Methylisorhamnetin	>4	>4	>4	>4	2	1	>4	>4
3-*O*-Methylgalangin	1	0.5	0.5	0.25	0.25	0.38	0.38	0.5
3,7-*O*-Dimethylgalangin	>4	>4	>4	>4	>4	>4	>4	>4

**Table 6 molecules-27-01149-t006:** Antibacterial activity of some selected flavonoids against several antibiotic resistant bacteria.

Flavonoids	MIC (μg/mL)		
	MRSA	VRE	*B. cepacia*
Luteolin	512	>512	>512
Datiscetin	512	>512	>512
Kaempferol	>512	>512	>512
Quercetin	256	512	>512
Myricetin	128	128	32

**Table 7 molecules-27-01149-t007:** Antibacterial activity of flavonoids isolated from *Psoralea corylifolia* seeds.

Flavonoids	MIC (mM)		Flavonoids	MIC (mM)	
	*S. aureus*	*S. epidermidis*		*S. aureus*	*S. epidermidis*
Corylifol A	0.147	0.147	Bavachin	0.037	0.037
Corylifol B	0.037	0.037	Bavachinin	0.018	0.018
Corylifol C	>0.147	>0.147	Corylin	>0.147	>0.147
Neobavaisoflavone	0.037	0.037	1-[2,4-dihydroxy-3-(2-hydroxy-3-methyl-3-butenyl)phenyl]-3-(4-hydroxyphenyl)-2-propen-1-one	>0.147	>0.147
Isobavachalcone	0.018	0.009	8-prenyldaidzein	>0.147	>0.147
7,8-dihydro-8-(4-hydroxyphenyl)-2,2-dimethyl-2H,6H-benzo[1,2-*b*:5,4-*b*′]dipyran-6-one	0.037	0.037	Bakuchalcone	>0.147	>0.147
Isoneobavaiso-flavone	0.073	0.037	Brosimacutin G	>0.147	>0.147
Bavachalcone	0.037	0.018	Erythrinin A	0.018	0.018
Bakuchiol (Control)	0.037	0.018	Magnolol (Control)	0.037	0.018

**Table 8 molecules-27-01149-t008:** Antibacterial activities of naturally occurring isoflavones **3a**–**o**.

Com. No.	R_1_	R_2_	R_3_	R_4_	R_5_	R_6_	MIC (μg/mL)					
							* B.e	* E.f	* L.m	* S.a	* S.e	* S.p
*** 3a**	H	OMe	OH	H	H	OMe	-	-	-	-	-	-
*** 3b**	H	OMe	OMe	H	H	OMe	-	-	-	-	-	-
*** 3c**	H	H	OMe	H	H	OMe	-	-	-	-	-	-
*** 3d**	OMe	H	H	OMe	H	OMe	-	-	-	-	-	-
*** 3e**	H	OMe	OH	H	H	H	-	-	-	-	-	-
*** 3f**	H	OH	OH	H	H	OH	32	-	128	16	32	64
*** 3g**	H	H	OH	H	OH	OH	-	128	128	-	-	-
*** 3h**	OH	H	OMe	H	H	OMe	-	-	-	-	-	-
*** 3i**	OH	H	OH	H	H	OMe	64	-	64	-	-	32
*** 3j**	H	H	OH	OH	H	OH	-	-	-	-	128	-
*** 3k**	OH	H	OMe	H	H	OH	-	-	-	-	-	-
*** 3l**	H	OMe	OH	H	H	OH	-	-	-	-	-	-
*** 3m**	H	H	OH	H	H	OH	-	-	-	-	-	-
*** 3n**	OH	H	OH	H	H	OH	128	-	-	-	-	64
*** 3o**	H	H	OH	H	H	OMe	-	-	-	-	-	-

* 6,4′-dimethoxy-7-hydroxyisoflavone (**3a**), 6,7,4′-trimethoxyisoflavone (**3b**), 7,4′-dimethoxyisoflavone (**3c**), 5,7,4′-trimethoxyisoflavone (**3d**), 7-hydroxy-6methoxyisoflavone (**3e**), 6,7,4′-trihydroxyisoflavone (**3f**), 7,3′,4′-trihydroxyisoflavone (**3g**), 7,4′-dimethoxy-5-hydroxyisoflavone (**3h**), 5,7-dihydroxy-4′-methoxyisoflavone (**3i**), 7,8,4′-trihydroxyisoflavone (**3j**), 5,4′-dihydroxy-7-methoxyisoflavone (**3k**), 7,4′-dihydroxy-6-methoxyisoflavone (**3l**), 7,4′-dihydroxyisoflavone (**3m**), 5,7,4′-trihydroxyisoflavone (**3n**), and 7-hydroxy-4′-methoxyisoflavone (**3o**); * B.c., *B. cereus*; E.f., *E.s faecalis*; L.m., *L. monocytogenes*; S.a., *S. aureus*; S.e., *S. epidermidis*; S.p., *Streptococcus pyogenes*.

**Table 9 molecules-27-01149-t009:** Chalcones (**4a**–**k**) showed good antibacterial activity against Gram-positive bacteria.

Com. No.	R_1_	R_2_	R_3_	R_4_	MIC (μg/mL)
*M. tuberculosis*	*E. faecalis*	MSSA	MRSA	*E. coli* (K12)	*E. coli* (ΔtolC)
*** 4a**	OH	OH	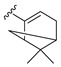	-	100	3.13	1.56	1.56	>200	1.56
*** 4b**	MOMO	OH	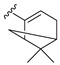		100	>200	>200	>200	>200	25
*** 4c**	OH	OH	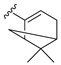	OH	>200	25	>200	200	>200	6.25
*** 4d**	OH	OH	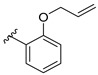	-	200	12.5	6.25	0.39	>200	3.13
*** 4e**	OH	OH	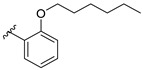	-	>200	1.56	3.13	0.78	>200	>200
*** 4f**	OH	OH	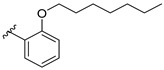	-	>200	1.56	25	6.25	>200	>200
*** 4g**	OH	OH	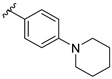	-	100	25	3.13	3.13	>200	12.5
*** 4h**	OH	OH	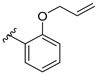	OH	>200	50	>200	25	>200	12.5
*** 4i**	OH	-	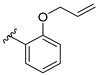	-	200	100	12.5	6.25	>200	6.25
*** 4j**	OH	OH	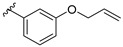	-	50	>200	6.25	12.5	>200	3.13
*** 4k**	OH	OH	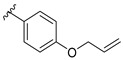	-	50	100	25	>200	>200	>200

* (*E*)-1-(2,4-Dihydroxyphenyl)-3-(6,6-dimethylbicyclo[3.1.1]-hept-2-en-2-yl)prop-2-en-1-one (**4a**), (*E*)-3-(6,6-Dimethylbicyclo[3.1.1]hept-2-en-2-yl)-1-(2-hydroxy-4-(methoxymethoxy)phenyl)prop-2-en-1-one (**4b**), (*E*)-3-(6,6-Dimethylbicyclo[3.1.1]hept-2-en-2-yl)-1-(2,4,6-trihydroxyphenyl)prop-2-en-1-one (**4c**), (*E*)-3-(2-(Allyloxy)phenyl)-1(2,4-dihydroxyphenyl)prop-2-en-1-one (**4d**), (*E*)-1-(2,4-Dihydroxyphenyl)-3-(2-(hexyloxy)phenyl)prop-2-en-1-one (**4e**), (*E*)-1-(2,4-Dihydroxyphenyl)-3-(2-(octyloxy)phenyl)prop-2-en-1-one (**4f**), (*E*)-1-(2,4-Dihydroxyphenyl)-3-(4-(piperidin-1-yl)phenyl)prop-2-en-1-one (**4g**), (*E*)-3-(2-(Allyloxy)phenyl)-1-(2,4,6-trihydroxyphenyl)prop-2-en-1-one (**4h**), (*E*)-3-(2-(Allyloxy)phenyl)-1-(4-hydroxyphenyl)prop-2-en-1-one (**4i**), (*E*)-3-(3-(Allyloxy)phenyl)-1-(2,4-dihydroxyphenyl)prop-2-en-1-one (**4j**), and (*E*)-3-(4-(Allyloxy)phenyl)-1-(2,4-dihydroxyphenyl)prop-2-en-1-one (**4k**).

**Table 10 molecules-27-01149-t010:** Anti-MRSA activities of flavonoids isolated from Kenyan plants.

Com. No.	R_1_	R_2_	R_3_	R_4_	MIC (μg/mL)			
					MRSA3	MRSA4	MRSA6	MRSA8
*** 5a**	OH	OH	H	OMe	-	64	128	-
*** 5b**	OMe	OH	OMe	OH	128	16	32	64
*** 5c**	OCOMe	OMe	H	OCOMe	-	128	-	-

* 3′,5′-dihydroxy-1′-methoxychalcone (**5a**), 1′,3′-dihydroxy-2′,5′-dimethoxychalcone (**5b**), 1,5-diacetate-3′-methoxychalcone (**5c**).

**Table 11 molecules-27-01149-t011:** Antibacterial activities of isobavachalcone and α-mangostin against multidrug resistant bacterial strains.

Compounds	MIC (μg/mL)							
	*S. aureus* (ATCC 29213)	MRSA (T144)	MRSA (65322)	*E. faecium*	VRE (CAU 383)	VRE (CAU 369)	*E. coli* (ATCC 25922)	*E. coli* (B2)
α-mangostin	1	1	0.5	0.5	0.5	2	>128	>128
Isobavachalcone	4	4	4	8	4	1	>128	>128
Vancomycin	1	0.5	1	>128	>128	>128	>128	>128

**Table 12 molecules-27-01149-t012:** Antibacterial activities of novel sulfur-containing tricyclic flavonoids (**6a**–**h**) against *S. aureus* and *E. coli*.

Com. No.	R_1_	R_2_	R_3_	MIC (μg/mL)	
				*S. aureus*	*E. coli*
*** 6a**	H	Br	Cl	0.48	3.9
*** 6b**	H	Br	OMe	1.95	125
*** 6c**	I	I	Cl	0.97	15.62
*** 6d**	Br	Br	Cl	1.95	7.81
*** 6e**	H	H	Cl	1.95	62.5

* 8-bromo-4-(4-chlorophenyl)-*N*,*N*-diethyl-4*H*-[1,3]dithiolo[4,5-*c*]chromen-2-amine (**6a**), 8-bromo-*N*,*N*-diethyl-4-(4-methoxyphenyl)-4*H*-[1,3]dithiolo[4,5-*c*]chromen-2-amine (**6b**), 4-(4-chlorophenyl)-*N*,*N*-diethyl-6,8-diiodo-4*H*-[1,3]dithiolo[4,5-*c*]chromen-2-amine (**6c**), 6,8-dibromo-4-(4-chlorophenyl)-*N*,*N*-diiethyl-4*H*-[1,3]dithiolo[4,5-*c*]chromen-2-amine (**6d**), and 4-(4-chlorophenyl)-*N*,*N*-diethyl-4H-[1,3]dithiolo[4,5-*c*]chromen-2-amine (**6e**).

**Table 13 molecules-27-01149-t013:** Sulfur-containing tricyclic flavonoids (**7a**–**m**) as antibacterial agent.

Com. No.	R_1_	R_2_	R_3_	MIC (μg/mL)		Com. No.	R_1_	R_2_	R_3_	MIC (μg/mL)	
				*S. aureus*	*E. coli*					*S. aureus*	*E. coli*
*** 7a**	Br	NMe_2_	Cl	7.81	15.62	*** 7i**	I	NEt_2_	Cl	0.48	3.9
*** 7b**	Br	Pyrrolidine	Cl	1.95	3.90	*** 7j**	I	NEt_2_	Br	0.48	3.9
*** 7c**	Br	Piperidine	Cl	62.5	125	*** 7k**	I	NEt_2_	I	0.48	3.9
*** 7d**	Br	NEt_2_	F	1.95	15.62	*** 7l**	I	NEt_2_	H	0.97	7.81
*** 7e**	Br	NEt_2_	Br	0.48	3.9	*** 7m**	H	NEt_2_	H	62.5	62.5
*** 7f**	Br	NEt_2_	I	0.48	3.9	Kanamycin	-	-	-	1.95	7.81
*** 7g**	Br	NEt_2_	H	1.95	7.81	Ampicillin	-	-	-	7.81	7.81
*** 7h**	I	NEt_2_	F	1.95	7.81						

* 2-*N*,*N*-Dimethylamino-8-bromo-4-(4-chlorophenyl)-4*H*-1,3-dithiol[4,5-*c*]chromen-2-ylium tetrafluoroborate (**7a**), 2-(Pyrrolidin-1-yl)-8-bromo-4-(4-chlorophenyl)-4*H*-1,3-dithiol[4,5-*c*]chromen-2-ylium tetrafluoroborate (**7b**), 2-(Piperidin-1-yl)-8-bromo-4-(4-chlorophenyl)-4*H*-1,3-dithiol[4,5-*c*]chromen-2-ylium tetrafluoroborate (**7c**), 2-*N*,*N*-Diethylamino-8-bromo-4-(4-fluorophenyl)-4*H*-1,3-dithiol[4,5-*c*]chromen-2-ylium tetrafluoroborate (**7d**), 2-*N*,*N*-Diethylamino-8-bromo-4-(4-bromophenyl)-4*H*-1,3-dithiol[4,5-*c*]chromen-2-ylium tetrafluoroborate (**7e**), 2-*N*,*N*-Diethylamino-8-bromo-4-(4-iodophenyl)-4*H*-1,3-dithiol[4,5-*c*]chromen-2-ylium tetrafluoroborate (**7f**), 2-*N*,*N*-Diethylamino-8-bromo-4-phenyl-4*H*-1,3-dithiol[4,5-*c*]chromen-2-ylium tetrafluoroborate (**7g**), 2-*N*,*N*-Diethylamino-8-iodo-4-(4-fluorophenyl)-4*H*-1,3-dithiol[4,5-*c*]chromen-2-ylium tetrafluoroborate (**7h**), 2-*N*,*N*-Diethylamino-8-iodo-4-(4-chlorophenyl)-4*H*-1,3-dithiol[4,5-*c*]chromen-2-ylium tetrafluoroborate (**7i**), 2-*N*,*N*-Diethylamino-8-iodo-4-(4-bromophenyl)-4*H*-1,3-dithiol[4,5-*c*]chromen-2-ylium tetrafluoroborate (**7j**), 2-*N*,*N*-Diethylamino-8-iodo-4-(4-iodophenyl)-4*H*-1,3-dithiol[4,5-*c*]chromen-2-ylium tetrafluoroborate (**7k**), 2-*N*,*N*-Diethylamino-8-iodo-4-phenyl-4*H*-1,3-dithiol[4,5-*c*]chromen-2-ylium tetrafluoroborate (**7l**), and 2-*N*,*N*-Diethylamino-4-phenyl-4*H*-1,3-dithiol[4,5-*c*]chromen-2-ylium tetrafluoroborate (**7m**).

**Table 14 molecules-27-01149-t014:** Antibacterial activity of chalcones (**8a**–**c**) against *S. aureus*.

Com. No.	R	Antibacterial Action against *S. aureus*	
		Inhibition Zone (mm)	MIC (ppm)
*** 8a**	H	11	92
*** 8b**	F	13	88
*** 8c**	Cl	9	93
Ampicillin	-	11	-

* 4-((*E*)-3-Oxo-3-phenylprop-1-enyl)benzoic acid (**8a**), 4-((*E*)-3-(4-Fluorophenyl)-3-oxoprop-1-enyl)benzoic acid (**8b**), and 4-((*E*)-3-(4-Chlorophenyl)-3-oxoprop-1-enyl)benzoic acid (**8c**).

**Table 15 molecules-27-01149-t015:** Antibacterial activity of the substituted chalcones contaning nitor, amino, hydroxyl, methoxy, and chloro groups (**9a**–**n**).

Com. No.	R_1_	R_2_	R_3_	R_4_	R_5_	R_6_	MIC (μg/mL)				
							*S. aureus*	*B. subtilis*	*E. coli*	*P. aeruginosa*	*S. enterica*
*** 9a**	NO_2_	H	H	OH	H	H	4.64	4.64	2.32	1.16	2.32
*** 9b**	NO_2_	H	NO_2_	H	H	H	16.76	4.19	1.05	2.10	4.19
*** 9c**	NO_2_	H	OMe	OMe	OMe	H	3.64	1.82	1.82	1.82	3.64
*** 9d**	NO_2_	NO_2_	H	H	H	H	2.10	4.19	2.10	2.10	4.19
*** 9e**	NO_2_	H	OEt	OH	H	H	1.00	3.99	3.99	1.99	1.99
*** 9f**	NO_2_	H	OMe	OH	H	H	4.18	2.09	2.09	2.09	4.18
*** 9g**	NO_2_	H	H	N(Et)_2_	H	H	3.85	3.85	1.93	1.93	1.93
*** 9h**	NO_2_	H	H	Br	H	H	3.76	3.76	0.94	1.88	1.88
*** 9i**	NO_2_	H	H	NO_2_	H	H	4.19	4.19	2.10	2.10	4.19
*** 9j**	NO_2_	H	OMe	OMe	H	H	1.00	1.99	1.00	1.99	3.99
*** 9k**	NO_2_	OMe	H	H	H	H	4.41	2.21	4.41	2.21	2.21
*** 9l**	NO_2_	OH	H	H	H	H	2.32	4.64	4.64	2.32	4.64
*** 9m**	NH_2_	H	H	Cl	H	H	2.43	4.85	4.85	2.43	2.43
*** 9n**	OH	H	OMe	OMe	OMe	H	1.99	3.98	3.98	1.99	3.98
Cefadroxil	-	-	-	-	-	-	1.72	1.72	1.72	1.72	1.72

* (*E*)-3-(4-Hydroxyphenyl)-1-(4-Nitrophenyl)prop-2-en-1-one (**9a**), (*E*)-3-(3-Nitrophenyl)-1-(4-Nitrophenyl)prop-2-en-1-one (**9b**), (*E*)-3-(3,4,5-Trimethoxyphenyl)-1-(4-Nitrophenyl)Prop-2-en-1-one (**9c**), (*E*)-3-(2-Nitrophenyl)-1-(4-Nitrophenyl)prop-2-en-1-one (**9d**), (*E*)-3-(3-Ethoxy-4-hydroxyphenyl)-1-(4-nitrophenyl)prop-2-en-1-one (**9e**), (*E*)-3-(4-Hydroxy-3-methoxyphenyl)-1-(4-Nitrophenyl)prop-2-en-1-one (**9f**), (*E*)-3-(4-(Diethylamino)phenyl)-1-(4-Nitrophenyl)prop-2-en-1-one (**9g**), (*E*)-3-(4-Bromophenyl)-1-(4-Nitrophenyl)prop-2-en-1-one (**9h**), (*E*)-1,3-Bis(4-nitrophenyl)Prop-2-en-1-one (**9i**), (*E*)-3-(3,4-Dimethoxyphenyl)-1-(4-Nitrophenyl)prop-2-en-1-one (**9j**), (*E*)-3–(2-Methoxyphenyl)-1-(4–Nitrophenyl)prop-2-en-1-one (**9k**), (*E*)-3-(2–Hydroxyphenyl)-1-(4-Nitrophenyl)prop-2-en-1-one (**9l**), (*E*)-1-(4-Aminophenyl)-3-(4-Chlorophenyl)prop-2-en-1-one (**9m**), and (*E*)-1-(4-Hydroxyphenyl)-3-(3,4,5-Trimethoxyphenyl)prop-2-en-1-one (**9n**).

**Table 16 molecules-27-01149-t016:** Antibacterial activity of isoflavones and their derivatives from bark of *Erythrina lysistemon*.

Compounds	MIC (μg/mL)				
	*B. cereus*	*S. aureus*	*S. epidermidis*	*E. coli*	*P. aeruginosa*
Erybraedin A	1	2	2	2	20
Phaseollidin	10	10	5	20	20
Abyssinone V-4′ methyl ether	26	59	117	260	260
Eryzerin C	10	5	2	5	5
Alpumisoflavone	31	31	125	125	20
Cristacarpin	156	156	412	625	78
Lysisteisoflavone	2	62	26	6	31

**Table 17 molecules-27-01149-t017:** Antibacterial activity of flavonoids from *Artocarpus heterophyllus*.

Compounds	MIC (μg/mL)						
	*Streptococcus mutans*	*S. pyrogenes*	*B. subtilis*	*S. aureus*	*S. epidermidis*	*P. aeruginosa*	*E. coli*
Dihydromorin	31.25	15.62	62.5	62.5	31.25	-	-
Norartocarpetin	125.0	31.25	250.0	125.0	250.0	-	-
Ampicillin	0.50	0.50	0.50	0.50	0.50	31.25	0.25

**Table 18 molecules-27-01149-t018:** Antibacterial activity of carboxylated chalcones against *S. aureus*.

Com. No.	R	MIC (μM)	Com. No.	R	MIC (μM)	Com. No.	R	MIC (μM)	Com. No.	R	MIC (μM)
*** 10a**	H	>300	*** 10g**	4-OPh	-	*** 10m**	3-OPh	-	*** 10s**	3,5-Di-CF_3_	2
*** 10b**	4-CF_3_	150	*** 10h**	3-CF_3_	40	*** 10n**	2-CF_3_	300	*** 10t**	3,5-Di-Br	2
*** 10c**	4-Cl	150	*** 10i**	3-Br	75	*** 10o**	2-Br	150	*** 10u**	3,5-Di-Cl	40
*** 10d**	4-Me	>300	*** 10j**	3-Cl	75	*** 10p**	2-Cl	150	*** 10v**	3,5-Di-Me	75
*** 10e**	4-OMe	>300	*** 10k**	3-NO_2_	300	*** 10q**	2-OH	>300	*** 10w**	3,5-Di-F	150
*** 10f**	4-OH	>300	*** 10l**	3-OH	>300	*** 10r**	2-OBu	-	*** 10x**	3,5-Di-OMe	>300

Control drug: Ciprofloxacin (MIC = 2 μM) and Linezolid (MIC = 2 μM). * 4-(3-Phenyl-acryloyl)-benzoic acid (**10a**), 4′-Carboxy-4-trifluoromethyl-chalcone (**10b**), 4′-Carboxy-4-chloro-chalcone (**10c**) 4′-Carboxy-4-methyl-chalcone (**10d**), 4′-Carboxy-4-methoxy-chalcone (**10e**), 4′-Carboxy-4-hydroxy-chalcone (**10f**), 4′-Carboxy-4-phenoxy-chalcone (**10g**), 4′-Carboxy-3-trifluoromethyl-chalcone (**10h**), 4′-Carboxy-3-bromo-chalcone (**10i**), 4′-Carboxy-3-chloro-chalcone (**10j**), 4′-Carboxy-3-nitro-chalcone (**10k**), 4′-Carboxy-3-hydroxy-chalcone (**10l**), 4′-Carboxy-3-phenoxy-chalcone (**10m**), 4′-Carboxy-2-trifluoromethyl-chalcone (**10n**), 4′-Carboxy-2-bromo-chalcone (**10o**), 4′-Carboxy-2-chloro-chalcone (**10p**), 4′-Carboxy-2-hydroxy-chalcone (**10q**), 4′-Carboxy-2-butoxy-chalcone (**10r**), 4′-Carboxy-3,5-bis-trifluoromethyl-chalcone (**10s**), 4′-Carboxy-3,5-dibromo-chalcone (**10t**), 4′-Carboxy-3,5-dichloro-chalcone (**10u**), 4′-Carboxy-3,5-dimethyl-chalcone (**10v**), 4′-Carboxy-3,5-difluoro-chalcone (**10w**), and 4′-Carboxy-3,5-dimethoxy-chalcone (**10x**).

**Table 19 molecules-27-01149-t019:** Antibacterial activities of tested chalcones against *B. bronchiseptica*.

Com. No.	R	(MIC: mg/mL)	Com. No	R	(MIC: mg/mL)	Com. No	R	(MIC: mg/mL)	Com. No	R	(MIC: mg/mL)
*** 11a**	H	0.5	*** 11g**	4-OMe	0.6	*** 11m**	2-Cl	0.5	*** 11s**	3-NO_2_	0.8
*** 11b**	2-OH	0.7	*** 11h**	3,4-OMe	-	*** 11n**	3-Cl	0.3	*** 11t**	4-NO_2_	1.0
*** 11c**	3-OH	0.7	*** 11i**	3-OH, 4-OMe	-	*** 11o**	4-Cl	0.2	*** 11u**	4-NMe_2_	-
*** 11d**	4-OH	0.8	*** 11j**	2-F	-	*** 11p**	3-Br	0.2	*** 11v**	4-Me	0.7
*** 11e**	2-OMe	0.5	*** 11k**	3-F	-	*** 11q**	4-Br	-	*** 11w**	2-Me, 3,4-OMe	-
*** 11f**	3-OMe	1.0	*** 11l**	4-F	0.4	*** 11r**	2-NO_2_	1.0			

* 1-(3′-Hydroxyphenyl)-3-(phenyl)-2-propen-1-one (**11a**), 1-(3′-Hydroxyphenyl)-3-(2-hydroxyphenyl)-2-propen-1-one (**11b**), 1-(3′-Hydroxyphenyl)-3-(3-hydroxyphenyl)-2-propen-1-one (**11c**), 1-(3′-Hydroxyphenyl)-3-(4-hydroxyphenyl)-2-propen-1-one (**11d**), 1-(3′-Hydroxyphenyl)-3-(2-methoxyphenyl)-2-propen-1-one (**11e**), 1-(3′-Hydroxyphenyl)-3-(3-methoxyphenyl)-2-propen-1-one (**11f**), 1-(3′-Hydroxyphenyl)-3-(4-methoxyphenyl)-2-propen-1-one (**11g**), 1-(3′-Hydroxyphenyl)-3-(3,4-dimethoxyphenyl)-2-propen-1-one (**11h**), 1-(3′-Hydroxyphenyl)-3-(4-hydroxy,3-methoxyphenyl)-2-propen-1-one (**11i**), 1-(3′-Hydroxyphenyl)-2-(2-fluorophenyl)-2-propene-1-one (**11j**), 1-(3′-Hydroxyphenyl)-2-(3-fluorophenyl)-2-propene-1-one (**11k**), 1-(3′-Hydroxyphenyl)-2-(4-fluorophenyl)-2-propen-1-one (**11l**), 1-(3′-Hydroxyphenyl)-3-(2-chlorophenyl)-2-propen-1-one (**11m**), 1-(3′-Hydroxyphenyl)-3-(3-chlorophenyl)-2-propen-1-one (**11n**), 1-(3′-Hydroxyphenyl)-3-(4-chlorophenyl)-2-propen-1-one (**11o**), 1-(3′-Hydroxyphenyl)-2-(3-bromophenyl)-2-propen-1-one (**11p**), 1-(3′-Hydroxyphenyl)-2-(4-bromophenyl)-2-propen-1-one (**11q**), 1-(3′-Hydroxyphenyl)-3-(2-nitrophenyl)-2-propen-1-one (**11r**), 1-(3′-Hydroxyphenyl)-3-(3-nitrophenyl)-2-propen-1-one (**11s**), 1-(3′-Hydroxyphenyl)-3-(4-nitrophenyl)-2-propen-1-one (**11t**), 1-(3′-Hydroxyphenyl)-3-(4-*N*,*N*-dimethylphenyl)-2-propen-1-one (**11u**), and 1-(3′-Hydroxyphenyl)-3-(4-methylphenyl)-2-propen-1-one (**11v**), 1-(3′-Hydroxyphenyl)-3-(2-methyl-3,5-dimethoxyphenyl)-2-propen-1-one (**11w**).

**Table 20 molecules-27-01149-t020:** Antibacterial activities of flavonoids and their sulfonic derivatives against several bacterial pathogens.

Compounds	R_1_	R_2_	R_3_	MIC (mg/mL)					
				*E. coli*		*P. aeruginosa*		*S. aureus*	
				ATCC 25922	Clinical Isolates	ATCC 27853	Clinical Isolates	ATCC 29213	Clinical Isolates
Quercetin	H	OH	H	62.5	62.5	62.5	62.5	62.5	62.5
Morin	OH	H	H	3.9	3.9	3.9	62.5	31.2	31.2
NaQSA	H	OH	SO_3_Na	1000.0	62.5	31.2	1000.0	3.9	31.2
NaMSA	OH	H	SO_3_Na	62.5	31.2	31.2	31.2	3.9	31.2

**Table 21 molecules-27-01149-t021:** Antibacterial activities of anthocyanins bearing fatty acid moiety in sugar portion.

Compounds	R	MIC (μg/mL)							
		*P. aeruginosa*		*E. coli*		*S. aureus*		*E. faecalis*	
		ATCC 27853	PA004	ATCC 25922	PA002	ATCC 29213	Sa1	ATCC 29212	S007
*** 12a**	OH	>512	>512	>512	>512	>512	>512	>512	>512
*** 12b**	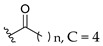	>512	>512	>512	>512	>512	>512	>512	>512
*** 12c**	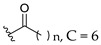	>512	128	8	4	>512	>512	>512	>512
*** 12d**	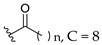	256	32	2	2	128	>512	8	8
*** 12e**	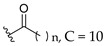	128	16	2	1	64	512	8	4
*** 12f**	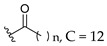	>512	>512	>512	>512	>512	>512	>512	>512

* Cyanidin-3-*O*-glucoside (**12a**), cyanidin-3-*O*-glucoside-C4 (**12b**), cyanidin-3-*O*-glucoside-C6 (**12c**), cyanidin-3-*O*-glucoside-C8 (**12d**), cyanidin-3-*O*-glucoside-C10 (**12e**), and cyanidin-3-*O*-glucoside-C12 (**12f**).

**Table 22 molecules-27-01149-t022:** Antibacterial activity of flavone derivatives against *B. subtitlis*, *S. aureus*, and *P. aeruginosa*.

Com. No.	R_1_	R_2_	MIC (μg/mL)		
			*B. subtilis*	*S. aureus*	*P. aeruginosa*
*** 13a**	H	H	12.5	25	25
*** 13b**	OMe	H	50	50	37.5
*** 13c**	H	Br	12.5	6.25	6.25
Ciprofloxacin	-	-	6.25	6.25	6.25

* 2-(4-(trifluoromethyl)phenyl)-4*H*-chromen-4-one (**13a**), 2-(4-(trifluoromethyl)phenyl)-7-methoxy-4H-chromen-4-one (**13b**), and 6-bromo-2-(4-(trifluoromethyl)phenyl)-4H-chromen-4-one (**13c**).

**Table 23 molecules-27-01149-t023:** Important structural characteristics of different classes of flavonoids for antibacterial activity.

**Chalcones**											
**References**	**Ring A**				**Ring B**				**Ring C**		
	**C5**	**C6**	**C7**	**C8**	**C2′**	**C3′**	**C4′**	**C5′**	**C2**	**C3**	**C4**
[77]	OH ↑		OH ↑						C2=C3 ↑		
[78]						OH ↑	OH ↑				
[79]					OH ↑ OMe ↓				Broken ring C ↑		C=O ↑
[82]	OH ↑		OH ↑						*O*-alkyl chain ↑		
[84]			OH ↑	Oxygenation ↑							
[65]		Prenyl ↑									
[92]			Halogen ↑				COOH ↑				
[94]							Bromine ↑				
[97]					Lipophilic subtituent ↑						
[98]					Lipophilic subtituent ↑						
**Flavanones**											
**References**	**Ring A**				**Ring B**				**Ring C**		
	**C5**	**C6**	**C7**	**C8**	**C2′**	**C3′**	**C4′**	**C5′**	**C2**	**C3**	**C4**
[71]	OH ↑	Geranyl ↑				OMe ↓	OMe ↓			OMe ↓	
[76]	Two hydroxyl groups ↑				Absence of hydroxyl groups ↑						
[79]	OH ↑ OMe ↓										C=O ↑
[80]	OH ↑		OH ↑		Trihydroxylation ↑ Glycosylation of hydroxyl groups ↓				Saturation of C2=C3 ↑		
[85]									Saturation of C2=C3 ↑		
[86]							Glycosyl ↑				
[96]									C2=C3 ↓ OH ↑		
**Flavones**											
**References**	**Ring A**				**Ring B**				**Ring C**		
	**C5**	**C6**	**C7**	**C8**	**C2′**	**C3′**	**C4′**	**C5′**	**C2**	**C3**	**C4**
[62]	OH ↑ OMe ↓	OH ↓ OMe ↑		OMe ↑		OH ↓ OMe ↓	OH ↑	OH ↓		OH ↓	
[64]						OH ↑	OH ↑	OH ↑	C2=C3 ↑	OH ↑	
[73]				OMe ↑						OH ↑	
[74]	OH ↑	OH ↑	OH ↑			OH ↑	OH ↑				
[75]						OMe ↑	OMe ↑ NO_2_ ↓ F ↑	NO_2_ ↓			
[76]	Two hydroxyl groups ↑				Absence of hydroxyl groups ↑				C2=C3 ↑		
[78]		Prenyl ↑	OMe ↑								
[79]	OH ↑ OMe ↓										
[80]	OH ↑		OH ↑		Trihydroxylation ↑ Glycosylation of hydroxyl groups ↓						
[100]	OH ↑		OH ↑	OH ↑		Sulfo ↑					
[102]	Halogen ↑						CF_3_ ↑				
[104]	OH ↑	OH ↓	OH ↑	OMe ↓					C2=C3 ↑	OH ↑	
**Isoflavones**											
**References**	**Ring A**				**Ring B**				**Ring C**		
	**C5**	**C6**	**C7**	**C8**	**C2′**	**C3′**	**C4′**	**C5′**	**C2**	**C3**	**C4**
[73]			Glycosyl ↓	Glycosyl ↓							
[81]	OH ↑ OMe ↓	OH ↑ OMe ↓	OH ↑ OMe ↓	OMe ↓			OH ↑ OMe ↓				
[95]	Prenyl ↑ OH ↑ OMe ↓				Prenyl ↑ OH ↑ OMe ↓						
[103]	OH ↑	Prenyl ↑									
**Flavan-3-ols**											
**References**	**Ring A**				**Ring B**				**Ring C**		
	**C5**	**C6**	**C7**	**C8**	**C2′**	**C3′**	**C4′**	**C5′**	**C2**	**C3**	**C4**
[70]	OH ↑		OH ↑				OH ↑			Long acyl chain ↑	
[73]										OH ↑	
**Anthocyanidins**											
**References**	**Ring A**				**Ring B**				**Ring C**		
	**C5**	**C6**	**C7**	**C8**	**C2′**	**C3′**	**C4′**	**C5′**	**C2**	**C3**	**C4**
[101]										Glycosyl ↑	
[105]	Electron withdrawing group ↑										

* ↑ indicates the substitution of mentioned moieties can increase flavonoids’ antibacterial activities. ↓ indicates the substitution of mentioned groups can reduce flavonoids’ antibacterial activities.

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
