# Peer review of "Antibacterial Effects of Flavonoids and Their Structure-Activity Relationship Study: A Comparative Interpretation"

_molecules, 2022, doi:10.3390/molecules27041149_

Round 1
Reviewer 1 Report
The Authors have introduced a lot of new data and tried to compared results of different studies in a Table. However, an in-depth critical discussion is yet to be done.
Author Response
Dear respected Sir
Tons of thank you for your thoughtful, helpful, and most kind review of our manuscript (molecules-1482299). Your comments and suggestions have been incorporated as appropriate into the revised draft and highlighted in yellow colour. Specific revisions are noted below.
Comment:
The Authors have introduced a lot of new data and tried to compared results of different studies in a Table. However, an in-depth critical discussion is yet to be done.
Authors' Response:
An in-depth critical discussion has been done to our utmost best as per the positive comments made by the reviewers. Hope the explanation is informative.
Please also see the attached file containing full revised article.

Reviewer 2 Report
The manuscript “Antibacterial Effects of Flavonoids and Their Structure-Activity 2 Relationship Study: A Comparative Interpretation” reports an interesting review about the antimicrobial activities and the specific features that flavonoids must fulfill in order to ensure a high antimicrobial activity.
In my opinion this manuscript is suitable to be published on Molecules as a contribution for the section: Natural Products Chemistry, Biosynthesis and Biological Activities of Flavonoids.
My main concern is related with the structure of the review. In my opinion the kind of information showed on “Discussion section” could be better organized. Which are the criteria used by authors to establish this order? Maybe it could make more sense if authors organize the information in different subsection according to the kind of flavonoids studied. This could allow authors to better compare the research performed by different authors using the same flavonoid family and to establish interconnection between them. I think that this kind of organization could be very interesting for this review and will help readers to focus on the special features that should fulfill each family of flavonoids to get a higher antimicrobial activity. For instance, it has no sense that Nielsen et al. 2004 (line 549) and Farooq et al. 2020 (line 494) that have studied the same kind of compounds, were described in different paragraphs and no interconnection between them is performed.
Another minor concern:
Line 69-70. “These compounds frequently display hydroxylation at positions 5 and 7 and the ring B and oxidation at the 4ʹ, 3ʹ,4ʹ or 3ʹ,4ʹ,5ʹ- positions due to their biosynthesis routes (46,54,55)” Better to say: “These compounds frequently display hydroxylation at positions 5 and 7 at A ring and oxidation at the 4ʹ; 3ʹ,4ʹ or 3ʹ,4ʹ,5ʹ positions at B ring due to their biosynthesis routes (46,54,55)”
In my opinion section 2 could be removed.
In table 1, please include as a footnote the meaning of the abbreviation used (OM481 OM505 OM584 OM623 COL N315 209P) as you have done at Table 2.
Line 426: bioflavonoids? Maybe authors mean biflavonoids.
Figure 19. Please correct the format for the legend.
In my opinion the introduction of a summary scheme with the chemical structure of all compounds that appear on the review could be very useful for readers. Or maybe instead of a global scheme, authors could include a summary scheme of all the structures that are commented on each subsection.
Please revise these two recent articles that perform antimicrobial analyses of natural flavonoids and derivatives.
Ortega-Vidal, J.; Cobo, A.; Ortega-Morente, E.; Gálvez, A.; Martínez-Bailén, M.; Salido, S.; Altarejos, J. Antimicrobial activity of phenolics isolated from the pruning wood residue of European plum (Prunus domestica L.). Ind. Crops Prod. 176, 2022,114296. https://doi.org/10.1016/j.indcrop.2021.114296.
Author Response
Dear respected Reviewer
Tons of thank you for your thoughtful, helpful, and most kind review of our manuscript (molecules-1482299). Your comments and suggestions have been incorporated as appropriate into the revised draft and highlighted in yellow colour. Specific revisions are noted below.
Comment 1: In my opinion the kind of information showed on “Discussion section” could be better organized. Which are the criteria used by authors to establish this order? Maybe it could make more sense if authors organize the information in different subsection according to the kind of flavonoids studied
Authors’ Response to Comment 1: Done and improved accordingly. The SAR information according to the kind of flavonoids studied have been added and rearranged at the end of discussion part to make it more informative and meaningful. A summary scheme of all the structures that are commented has been added at the end of discussion.
Comment 2: Line 69-70. “These compounds frequently display hydroxylation at positions 5 and 7 and the ring B and oxidation at the 4ʹ, 3ʹ,4ʹ or 3ʹ,4ʹ,5ʹ- positions due to their biosynthesis routes (46,54,55)” Better to say: “These compounds frequently display hydroxylation at positions 5 and 7 at A ring and oxidation at the 4ʹ; 3ʹ,4ʹ or 3ʹ,4ʹ,5ʹ positions at B ring due to their biosynthesis routes (46,54,55)”
Authors’ Response to Comment 2: Done and rectified accordingly.
Comment 3: In my opinion section 2 could be removed.
Authors’ Response to Comment 3: Thanks. The previously reviewers requested to add this information in our manuscript.
Comment 4: Line 426: bioflavonoids? Maybe authors mean biflavonoids.
Authors’ Response to Comment 4: This has been corrected accordingly.
Comment 5: Figure 19. Please correct the format for the legend.
Authors’ Response to Comment 5: Done, but I think there is some problem with this word document, since it reverts to unformatted one.
Comment 6: Please revise these two recent articles that perform antimicrobial analyses of natural flavonoids and derivatives. Ortega-Vidal, J.; Cobo, A.; Ortega-Morente, E.; Gálvez, A.; Martínez-Bailén, M.; Salido, S.; Altarejos, J. Antimicrobial activity of phenolics isolated from the pruning wood residue of European plum (Prunus domestica L.). Ind. Crops Prod. 176, 2022,114296. https://doi.org/10.1016/j.indcrop.2021.114296.
Authors’ Response to Comment 6: Both articles have been added and revised accordingly.

Round 2
Reviewer 1 Report
The Authors have tried and improved the article but the discussion yet lacks reasonable structure. Tables such as "Table 2. Isolated compounds from Paulownia tomentosa were showed to display potent antibacterial activity against several Gram-positive strains." or "Table 7. Antibacterial activity of flavonoids isolated from Psoralea corylifolia seeds." are inappropriate in such a review.
Author Response
Dear Respected Sir
We have incorporated all the corrections (highlighted with a green colour) just according to your constructive comments to make our review more informative. We are extremely grateful to you for helping us to further improve our review. We have added more structures to make it more meaningful. Thanks again
Sincerely yours
Corresponding author Qamar Uddin Ahmed

Reviewer 2 Report
Authors have checked and incorporated all the suggestions. In my opinion changes performed allow a better comprehension of this important topic and help readers to establish interconnection between the studies performed by different authors.
In my opinion this work is suitable to be published
Author Response
Thank you very much sir
This manuscript is a resubmission of an earlier submission. The following is a list of the peer review reports and author responses from that submission.
Round 1
Reviewer 1 Report
The paper titled "Antibacterial Effects of Flavonoids and Their Structure-Activity Relationship Study: A Comparative Interpretation" has a good idea behind it - but the execution is not good.
The sentence from paper "Hence, the latest information on the antibacterial activity of flavonoids is summarised in this review, with a special attention on the structure–activity relationship of this broad class of natural compounds to discover safe and potent antibacterial agents as natural products." describes it best.
However there is a big problem with it:
1) It has only 28 references.
2) References 1 and 2 are general information sources, and references 6 (2019.), 7 (2020.), and 15 (2019.) are quite quite similar reviews. So that comes to only 23 references.
3) But Review references 6, 7, and 15 have already reviewed references used in this paper namely: 4, 5, 9, 10, 12, 14, 18, 21, 24, 26, and 28. Which all comes to total of 12 references used which have not been mentioned before and of all those only one is from 2020 in period "after" reviews from point 2.
4) Under the Discussion part of paper, - there is no novel conclusions - only ones already made by authors cited put in one after another. Authors of this paper didn't go in-depth on various SAR information (other than already published) and real comparison of all papers reviewed is dreadfully missing.
5) Extremely generic conclusion of the whole "comparative study" is given in Conclusion section.
6) In the end it is only a summary that is left (look the sentence above) - no real novelty or discussion.
English language contains errors in grammar and style (marked yellow).
Some compound names should have proper style (marked yellow).
All the figures should be uniformed and consistent in size and font.
I cannot see the novelty of this paper in comparison to references used not to say that there are only a few and not so many recent ones.
Hence I must recommend the paper to be rejected or seriously revised.

Reviewer 2 Report
The manuscript entitled “Antibacterial Effects of Flavonoids and Their Structure-Activity Relationship Study: A Comparative Interpretation” aims to review the latest information on the antibacterial activity of flavonoids, especially on the structure-activity relationship of this class of natural compounds. This topic is of high importance for the humanity. The manuscript is, in general, well-written. Still, it is rather long, so it would be much easier to compartmentalize the Discussion part in order to make the manuscript more appealing for the reader. Also, I find the number of references too low for this kind of review.
Reviewer 3 Report
This manuscript is an attempt to perform a review on the antibacterial activity of flavonoids with a focus on their structure-activity relationship. The subject is important and such a review could be very useful. However, there are some serious concerns.
Major concerns:
- The Authors have not explained the methodology of collecting information for the review: what searches they performed, which databases they searched, etc.
- The review lacks any internal structure.
- There is no critical discussion of the literature data.
Minor concerns:
- Page 2, line 63 – 64: But what about chalcones? It is better to say "connected by a three-carbon unit, in most cases forming a heterocyclic… ” etc.
- Page 2: Move the two sentences in lines 68 – 70 to line 64, after "C6-C3-C6".
- 1: Correct: flavAnone, flavanonol
- 7: How is this synthetic scheme relevant to the subject of the review? The chemistry of flavonoid transformations is not the subject.